# A universal dual mechanism immunotherapy for the treatment of influenza virus infections

Xin Liu [1,2], Boning Zhang[1,2], Yingcai Wang[1,2], Hanan S. Haymour[1,2], Fenghua Zhang [1,2], Le-cun Xu[3], Madduri Srinivasarao [1,2] & Philip S. Low [1,2✉]

Seasonal influenza epidemics lead to 3–5 million severe infections and 290,000–650,000 annual global deaths. With deaths from the 1918 influenza pandemic estimated at >50,000,000 and future pandemics anticipated, the need for a potent influenza treatment is critical. In this study, we design and synthesize a bifunctional small molecule by conjugating the neuraminidase inhibitor, zanamivir, with the highly immunogenic hapten, dinitrophenyl (DNP), which specifically targets the surface of free virus and viral-infected cells. We show that this leads to simultaneous inhibition of virus release, and immune-mediated elimination of both free virus and virus-infected cells. Intranasal or intraperitoneal administration of a single dose of drug to mice infected with 100x $MLD_{50}$ virus is shown to eradicate advanced infections from representative strains of both influenza A and B viruses. Since treatments of severe infections remain effective up to three days post lethal inoculation, our approach may successfully treat infections refractory to current therapies.

[1] Department of Chemistry, Purdue University, West Lafayette, IN 47907, USA. [2] Purdue Institute for Drug Discovery, Purdue University, West Lafayette, IN 47907, USA. [3] Endocyte Inc., 3000 Kent Ave, West Lafayette, IN 47906, USA. ✉email: plow@purdue.edu

Estimates from the centers for disease control and prevention (CDC) report that between 9 and 45 million new cases of influenza occur each year in the USA, leading to 140,000–810,000 hospitalizations/year and 12,000–61,000 deaths/year[1]. The annual financial burden associated with treating these illnesses has been estimated at $4.6 billion and costs stemming from the accompanying loss of work have been projected at $7 billion[2]. While most influenza virus infections remain nonlethal and containable, their worldwide impact on morbidity and mortality remains one of the most adverse of any infectious disease.

Three general approaches have demonstrated efficacy in controlling influenza virus. First, vaccines against common influenza antigens have proven successful in limiting the severity and spread of the virus during years when the most aggressive viral strains are correctly predicted. Unfortunately, due to the rapid evolution of the virus, annual formulations of the vaccine often fail to match the most virulent strains, resulting in many vaccinated patients still contracting an infection[3–5]. Second, neuraminidase inhibitors have been designed to block the viral neuraminidases required for release of the virus from its host cell surface[6,7]. Although four neuraminidase inhibitors (zanamivir, oseltamivir, peramivir, and laninamivir) have been approved for treatment of influenza in different parts of the world, they commonly provide little benefit when administered more than two days after symptoms appear, leaving a large fraction of infected individuals with no treatment to mitigate symptoms[6]. Moreover, the emergence of low levels of variant viruses with reduced susceptibility to the above neuraminidase inhibitors has raised concerns that widespread circulation of viruses with reduced drug susceptibility can occur[8–10]. Third, baloxavir marboxil (Xofluza) and related drugs impede the synthesis of viral mRNAs by suppressing the cap-dependent endonuclease of both influenza A and B viruses[11,12]. However, although studies reveal that baloxavir can reduce the viral load and alleviate influenza symptoms, baloxavir-resistant strains have already been identified in patients[13], suggesting that its efficacy as a broad-spectrum therapy may be affected.

In the study below, we explore a hybrid of the above two therapeutic approaches, where we take advantage of both the potent antiviral activity of a broad spectrum viral neuraminidase inhibitor and the powerful immunological function of a vaccine. Briefly, we exploit the fact that influenza virus-infected cells express one or more viral proteins on their cell surfaces[14], thereby distinguishing them from adjacent healthy cells (Fig. 1a). While cell surface viral neuraminidases are intrinsically antigenic, we enhance their immunogenicity by decorating them with a potent hapten (i.e., in this case, a dinitrophenyl moiety; DNP) (Fig. 1b). Even though the origin of the antibodies against DNP and other nitroarenes is not known, they comprise ~1% of circulating antibodies in human serum and are competent to induce ligand-targeted cytotoxicity[15–18]. Moreover, rather than using a functionally inert ligand to target this hapten, we deliver the DNP moiety attached to a potent neuraminidase inhibitor, namely zanamivir (Fig. 1c), since zanamivir binds to neuraminidases of all known subtypes/lineages of influenza A and B viruses[19]. Because virtually all humans naturally express anti-DNP antibodies[18], the resulting zanamivir-DNP conjugate (zan-DNP) can not only block the activity of an essential viral enzyme, but also recruit the immune system to attack the virus or virus-infected cells.

Here, we show that zan-DNP inhibits neuraminidases of both influenza A and B viruses and concurrently recruits anti-DNP antibodies to virus-infected cells. Because this anti-DNP binding mediates destruction of the opsonized virus and virus-infected cells, treatment with zan-DNP is observed to eradicate even the severe viral infections where mice are inoculated with 100× $MLD_{50}$ of viral load and therapy is not initiated until 3 days post-infection. Considering that a single intranasal or intraperitoneal dose of zan-DNP yields a complete response, we suggest that further development of zan-DNP as a universal anti-influenza therapy is warranted.

## Results

**zan-DNP retains high binding affinity for neuraminidases.** In order to target a potent hapten to different subtypes/lineages of influenza virus and influenza virus-infected cells, we needed a receptor that was accessible on all viral particles and their infected host cells, as well as a ligand that would bind specifically to this influenza-specific receptor. Because influenza neuraminidase satisfies both criteria (i.e., it is abundantly expressed on both free viral particles and virus-infected cells), and since zanamivir binds to all known subtypes of groups A and B influenza virus neuraminidases[6], we selected zanamivir as the targeting ligand for delivery of a hapten to neuraminidase-expressing particles/cells. As shown in the crystal structure of a zanamivir-neuraminidase complex (Fig. 2a), the C-7 hydroxyl of zanamivir is exposed to solvent and available for linkage to an immunogenic hapten[20,21]. Based on this observation, we linked either a dinitrophenyl hapten (for therapeutic purposes) or rhodamine (for use in binding assays) to this C-7 hydroxyl via a PEG-related linker (Fig. 1c and Fig. 2b, respectively). As revealed in the binding curve of Fig. 2c, zanamivir-rhodamine conjugate (zan-rhodamine) associates with a representative member of group 1 neuraminidases (N1) of influenza A virus with a dissociation constant of 4.9 nM. That this interaction is neuraminidase-mediated is confirmed by the ability of 100-fold excess free zanamivir to block zan-rhodamine binding. Use of the same zan-rhodamine to measure binding of free zanamivir and zan-DNP via competitive binding analysis then yielded binding affinities of 0.3 nM and 0.8 nM, respectively. Moreover, use of the same rhodamine-labeled zanamivir conjugate to quantitate binding of zanamivir and zan-DNP to a representative member of group 2 neuraminidases (N2) of influenza A virus afforded similar binding constants of 1.0 nM and 1.1 nM, respectively (Fig. 2d). Because the substrate-binding sites are similar within all members of group 1 (N1, N4, N5, and N8) and group 2 (N2, N3, N6, N7, and N9) neuraminidases of influenza A virus[22], we conclude that attachment of a therapeutic or imaging agent to the C-7 position of zanamivir does not significantly interfere with zan-DNP binding to influenza virus A neuraminidases. The binding affinity of zan-DNP was also tested on two strains of influenza B virus (Fig. 2e, f). Even though attachment of DNP to zanamivir reduced the binding affinity of the conjugate for influenza B virus neuraminidases ~8 to 10-fold compared to zanamivir (i.e., a common consequence of ligand derivatization[23]), the binding affinity still remained in the low nanomolar range. Finally, an analogous binding study was performed on virus-infected normal human bronchial epithelial (NHBE) cells grown at air–liquid interface (ALI). As shown in Supplementary Fig. 1a, a similar binding constant of 1.4 nM was observed for zan-rhodamine associates with the neuraminidase (N1) of influenza A virus.

To assure that zan-DNP can also inhibit the neuraminidase activity necessary for protecting cells from virus infection, we compared the potencies of zanamivir and zan-DNP in suppressing the cytopathic effect caused by virus infection in an MDCK-influenza virus co-culture. As shown in Supplementary Fig. 2, zan-DNP can reduce the cytopathic effect of both influenza A (H1N1) and A(H3N2) virus infections with a potency similar to zanamivir alone ($EC_{50}$ = 1.7 and 7.6 nM, respectively). These data thus suggest that the use of zanamivir to target an attached

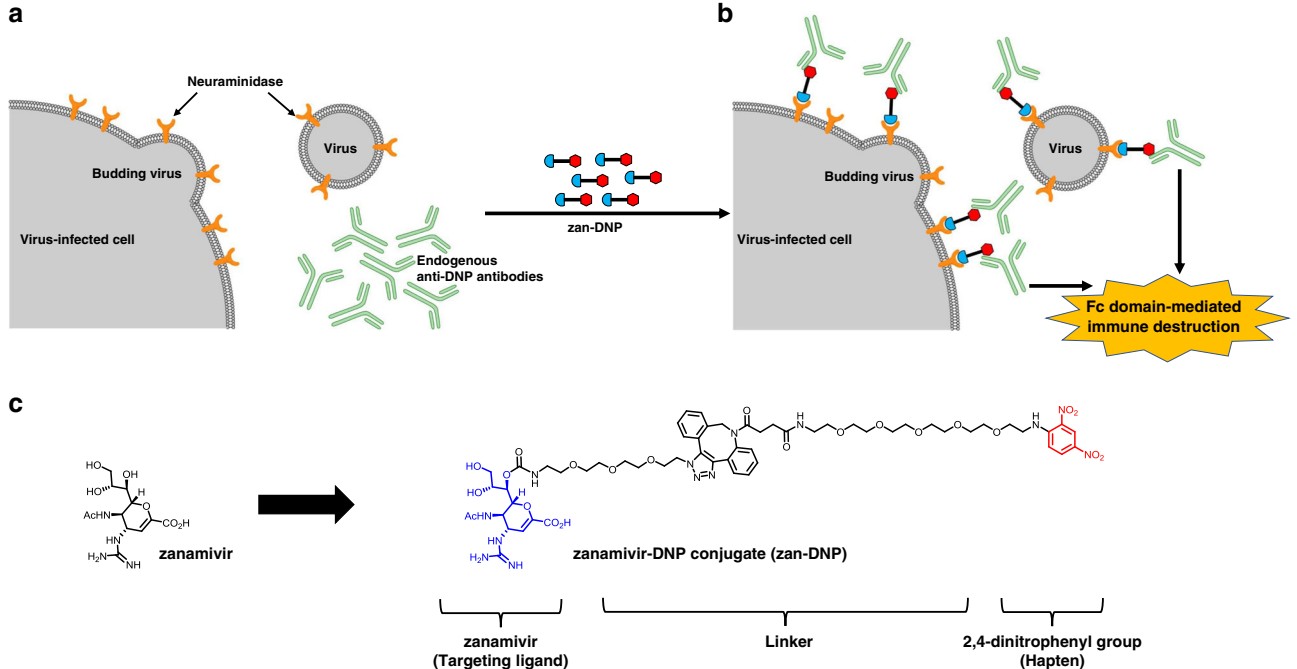

**Fig. 1 Dual mechanisms of action of neuraminidase-targeted anti-influenza therapy. a** Both influenza virus and influenza virus-infected cells express a viral neuraminidase (orange receptor) on their membrane surfaces. **b** Binding of a zanamivir-dinitrophenyl conjugate (zan-DNP) to these neuraminidases not only inhibits neuraminidase activity and thereby suppresses virus budding from the host cell, but also decorates both the free virus and infected host cell with the highly immunogenic hapten, dinitrophenyl (DNP). This painting of the virus and infected host cell with zan-DNP then recruits naturally occurring anti-DNP antibodies that are present in the blood of virtually all humans, resulting in opsonization of the virus/virus-infected cell and the consequent immune-mediated clearance of the virus/infected cell. **c** zan-DNP was synthesized by conjugating zanamivir (targeting ligand) with 2,4-dinitrophenyl group (hapten) via a polyethylene glycol (PEG) chain (linker).

immunogenic hapten to free virus or virus-infected cells will not significantly compromise the anti-neuraminidase activity of zanamivir.

**zan-DNP recruits anti-DNP antibodies to virus-infected cells.** The second therapeutic mechanism in our dual-function antiviral conjugate was designed to recruit the immune system to eradicate the virus/virus-infected cells. For this purpose, we decorated the virus/virus-infected cells with an immunogenic hapten that would recruit anti-hapten antibodies to the surface of the viral particles/ infected cells. Following evaluation of multiple haptens in the literature, the dinitrophenyl (DNP) hapten was selected for conjugation to zanamivir because anti-DNP antibodies are naturally present in essential all human sera; i.e., obviating the need to induce anti-DNP antibody production via vaccination against DNP[17,24].

To assess the ability of zan-DNP to recruit anti-DNP antibodies to virus-infected cells, we first employed biotinylated anti-DNP antibodies to enable visualization of the antibodies on virus-infected MDCK cells. As seen in the confocal micrographs of Fig. 3a, anti-DNP antibodies bind only to MDCK cells that have been infected with influenza A(H1N1) virus and then treated with zan-DNP. In contrast, no antibody binding was seen when MDCK cells were either not infected or infected with virus and treated with zan-DNP in the presence of 100-fold excess zanamivir (to block all zan-DNP binding to neuraminidase). Binding of anti-DNP antibodies could also be shown by quantitating the fluorescence of cell-bound streptavidin-PE in the presence of excess anti-DNP antibody as a function of the concentration of zan-DNP. As shown in Fig. 3b, as the concentration of zan-DNP was increased in the culture medium, binding of anti-DNP antibodies to infected MDCK cells steadily increased until all zan-DNP sites were eventually saturated.

Binding of anti-DNP antibodies to a representative member of group 2 neuraminidases of influenza A(H3N2) virus yielded very similar results (Fig. 3c). Thus, binding was totally dependent on the presence of zan-DNP, anti-DNP antibodies, and streptavidin-PE, and binding could be quantitatively prevented by blockade of the neuraminidase sites with excess zanamivir. Similarly, analysis of anti-DNP antibody binding as a function of zan-DNP concentration demonstrated saturable binding that was dependent on zan-DNP association with the infected MDCK cells (Fig. 3d). Finally, when an analogous antibody recruitment assay was performed on virus-infected NHBE cells grown at ALI, a similar dependence of antibody binding on the concurrent presence of zan-DNP, anti-DNP antibodies, and anti-human IgG-PE was observed (Supplementary Fig. 1b).

To document the ability of zan-DNP to recruit anti-DNP antibodies to virus-infected lung cells *in-vivo*, we challenged mice with 100x $MLD_{50}$ of A/California/07/2009 (H1N1)pdm09 virus prior to treatment with 0.5 μmol/kg zan-DNP or zanamivir plus anti-DNP antibodies (5 mg/kg, intravenous administration) (anti-DNP antibodies were added because mice naturally produce significantly lower levels of anti-DNP antibodies than humans, see Supplementary Fig. 3). Following digestion of the infected lungs, isolated lung cells were evaluated for anti-DNP antibody binding by flow cytometry. As shown in Fig. 3e and Supplementary Fig. 4, 24.4% of infected lung cells (i.e., hemagglutinin positive cells) were found to be opsonized with antibody in mice treated with zan-DNP. In contrast, replacement of zan-DNP with zanamivir resulted in only 0.8% of the infected lung cells exhibiting antibody positivity, and when uninfected mice were treated with zan-DNP only 1.7% of lung cells were found to be antibody positive. Finally, when zan-DNP was administered to virus-infected mice lacking anti-DNP antibodies, only 0.4% of infected lung cells were determined to be opsonized with

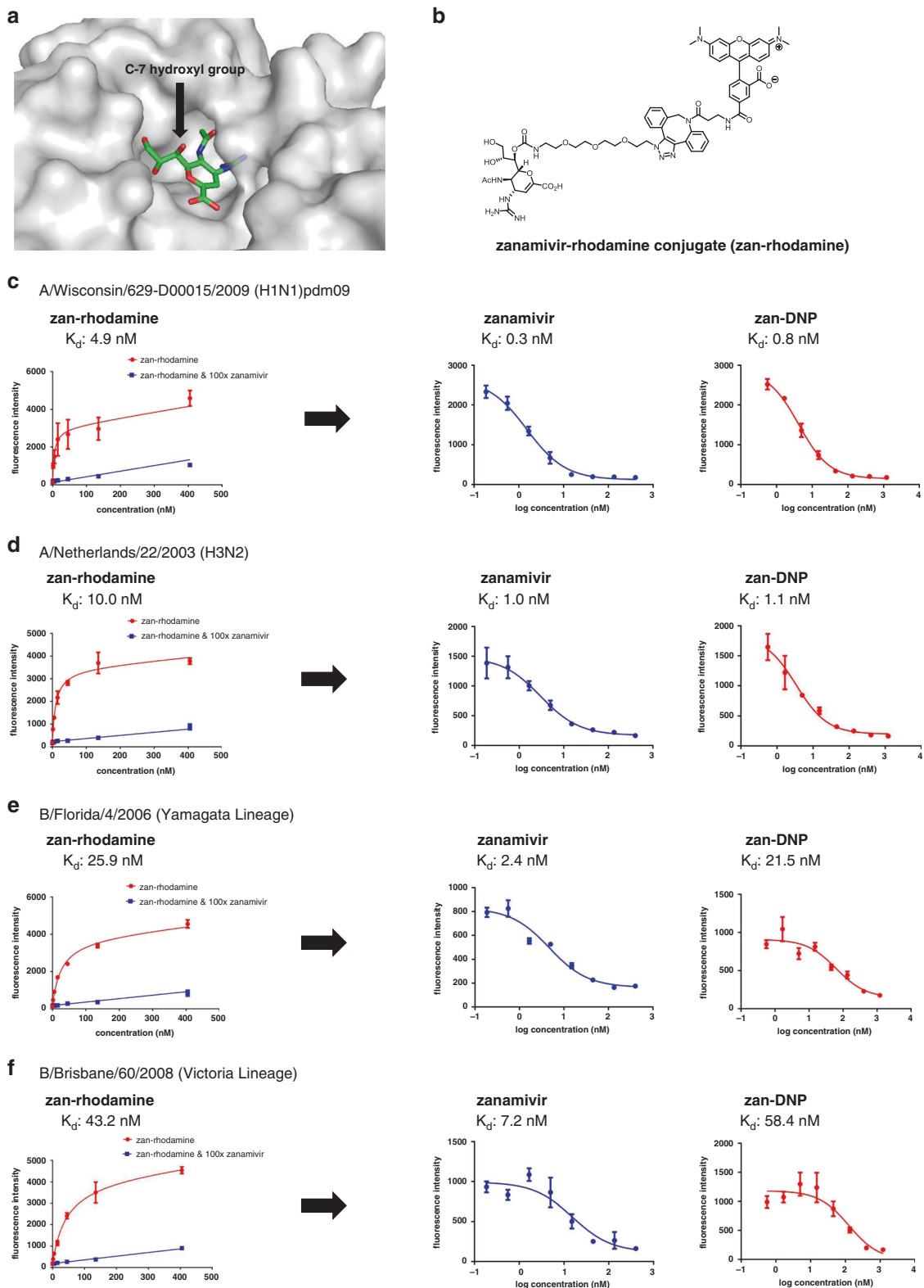

**Fig. 2 Binding affinity and specificity of zan-DNP for viral neuraminidases. a** Crystal structure of N1 neuraminidase of influenza A virus complexed with zanamivir (PDB code: 3TI5). Note that the C-7 hydroxyl of zanamivir is exposed to solvent and available for linkage to DNP. **b** Structure of zanamivir-rhodamine conjugate (zan-rhodamine). **c–f** Binding of zan-rhodamine (direct binding curve, left panel), zanamivir (competitive binding curve, central panel), and zan-DNP (competitive binding curve, right panel) to neuraminidase expressed on MDCK cells infected with influenza virus A/Wisconsin/629-D00015/2009 (H1N1)pdm09 (**c**), A/Netherlands/22/2003 (H3N2) (**d**), B/Florida/4/2006 (Yamagata Lineage) (**e**) or B/Brisbane/60/2008 (Victoria Lineage) (**f**). All the virus-infected cells were used 24 h post-infection for binding studies. Data are presented as mean values ± SD ($n = 3$).

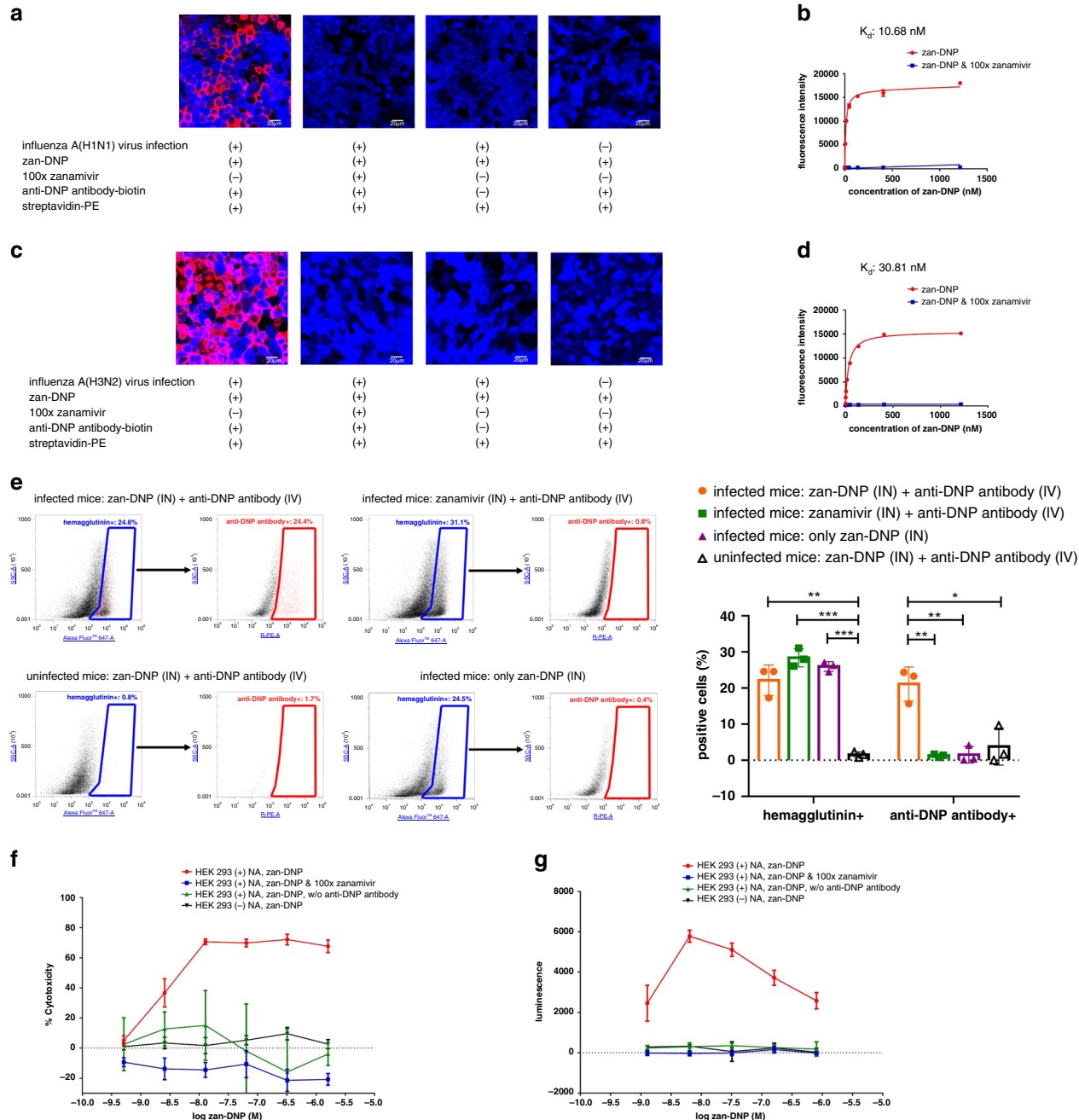

**Fig. 3 Anti-DNP antibodies recruiting, complement dependent cytotoxicity (CDC), and antibody-dependent cellular cytotoxicity (ADCC) of neuraminidase expressing cells induced by zan-DNP. a** and **c** Confocal micrographs of binding of biotinylated anti-DNP antibodies (visualized by streptavidin-PE) to the surface of A/Puerto Rico/8/1934 (H1N1) (**a**) and A/Aichi/2/1968 (H3N2) (**c**) virus-infected MDCK cells mediated by zan-DNP ($n = 3$). Nuclear staining is shown in blue, and streptavidin-PE staining is shown in red. **b, d** Binding of anti-DNP antibodies to influenza A(H1N1) (**b**) and A (H3N2) (**d**) virus-infected MDCK cells determined by quantitating the fluorescence of cell-bound streptavidin-PE in the presence of anti-DNP antibodies as a function of the concentration of zan-DNP ($n = 3$). All the virus-infected cells were used 24 h post-infection for the above studies. **e** Flowcytometry analyses of recruitment of anti-DNP antibodies to virus-infected cells mediated by zan-DNP in vivo ($n = 3$). A/California/07/2009 (H1N1)pdm09 virus-infected mice (3 mice/group) were first given zan-DNP and anti-DNP antibodies (3 days post-infection). 12 h later, the cells digested from the lung of virus-infected/uninfected mice were stained first with anti-hemagglutinin antibodies and then labeled with dye conjugated secondary antibodies against anti-hemagglutinin antibodies and anti-DNP antibodies. Left: selected flow cytometry plots; Right: summary graph. Statistical difference of antibody stained positive cells (%) between indicated two groups was analyzed by two-sided $t$ test (*$P < 0.05$, **$P < 0.01$, ***$P < 0.001$, see exact $P$ values in Supplementary Table 1). **f** CDC mediated killing of N1 neuraminidase transduced and nontransduced HEK 293 cells mediated by zan-DNP in the presence of anti-DNP antibodies and complement-preserved rabbit serum ($n = 3$). **g** Antibody-dependent cellular cytotoxicity (ADCC) of N1 neuraminidase transduced and nontransduced HEK 293 cells mediated by zan-DNP in the presence of anti-DNP antibodies and human FcγRIIIa expressing effector cells ($n = 3$). ADCC activity is proportional to the amount of firefly luciferase produced by effector cells (detected as the luminescence intensity). The coding sequence of N1 neuraminidase for the transduction of HEK 293 cells is from strain A/Puerto Rico/8/1934 (H1N1); Data are presented as mean values ± SD ($n = 3$).

antibody. These data demonstrate that significant antibody binding only occurs in vivo when lung cells are exposed to virus, zan-DNP and anti-DNP antibodies, and that absence of any one of these components abrogates antibody binding.

**zan-DNP mediates CDC and ADCC.** Having established that zan-DNP can recruit anti-DNP antibodies to the surfaces of virus-infected cells, we next elected to explore whether membrane bound anti-DNP antibodies could promote complement dependent cytotoxicity (CDC). For this purpose, HEK 293 cells were transduced with N1 neuraminidase of influenza A virus (to avoid any intrinsic cytotoxicity caused by the live virus, Supplementary Fig. 5 shows the binding affinity of zan-DNP to this transduced neuraminidase is 3.0 nM) and then incubated with zan-DNP plus anti-DNP antibodies and complement proteins. As shown in Fig. 3f, complement-mediated killing of the neuraminidase-expressing cell was dependent on the simultaneous presence of viral neuraminidase, zan-DNP and anti-DNP antibodies (more than 60% killing, red line), and abrogated (blue and black line) or significantly reduced (~10% killing, green line) when any one of these components was blocked or omitted. Importantly, maximum cell death was observed at a zan-DNP concentration of only 10 nM, emphasizing the potency of this immune contribution to virus elimination.

Second, to explore whether zan-DNP binding might mediate antibody-dependent cellular cytotoxicity (ADCC), N1-transduced HEK 293 cells were incubated with human anti-DNP antibodies plus Jurkat cells engineered to induce firefly luciferase expression upon participation in ADCC[25]. As seen in Fig. 3g, no ADCC was observed when (i) zan-DNP binding was blocked with 100-fold excess zanamivir (blue line), (ii) anti-DNP antibodies were deleted from the reaction mixture (green line), or (iii) HEK 293 cells not transduced with N1 neuraminidase were used as target cells (black line). However, potent ADCC was seen when N1-transduced cells were co-incubated with engineered Jurkat cells in the presence of both zan-DNP plus anti-DNP antibodies. These data together with the data on CDC demonstrate that zan-DNP can promote influenza virus-infected cell killing by multiple mechanisms, leading to likely resolution of the disease. Importantly, the bell-shaped response curves shown in Fig. 3g are anticipated, since high concentrations of zan-DNP will inhibit antibody binding after its bridging function has become saturated.

**zan-DNP is superior to zanamivir when tested in vivo.** With evidence that zan-DNP can promote destruction of neuraminidase expressing cells in the presence of anti-DNP antibodies, the question naturally arose whether zan-DNP can prevent influenza virus-infected mice from progressing to a lethal infection. To explore this issue, we first vaccinated BALB/c mice against DNP (because mice naturally produce significantly lower levels of anti-DNP antibodies than humans, see Supplementary Fig. 3) and then inoculated the mice intranasally (IN) with a lethal dose (100x $MLD_{50}$) of either influenza A(H1N1) or A (H3N2) virus (see Supplementary Fig. 6 for the general procedure of mouse therapy studies). Daily measurements of body weight then demonstrated that these severely infected mice began to lose weight 2 days post-infection and continued to lose weight until euthanasia was mandated due to weight loss ~6 days post-infection (Fig. 4a, left panel). To explore the ability of zanamivir (Relenza) to treat this severe infection, free zanamivir was administered IN twice daily for 5 days per its usual dosing regimen in humans[26]. As seen in Fig. 4a, treatment of mice twice daily with 0.5 µmol/kg (0.2 mg/kg) zanamivir beginning 24 h post-infection had little effect on body weight loss and only minor

impact on overall survival (blue lines), i.e., suggesting that 0.5 µmol/kg (0.2 mg/kg) zanamivir was ineffective in reversing a lethal viral infection (see Supplementary Fig. 7a for dose ranging study). Similarly, IN administered nontargeted DNP provided no protection against infection-induced weight loss or rapid mouse death (orange lines). In contrast, mice vaccinated against DNP and treated IN with zan-DNP were found to fully recover from the same lethal dose of influenza virus, as evidenced by prevention of virus-induced weight loss and virus-promoted death (red lines). That most of this protection was immune-mediated could be demonstrated by the observation that failure to vaccinate the mice against DNP abrogated the protection (purple lines). These data argue that zan-DNP is far more effective in treating a lethal influenza A(H1N1) virus infection than unmodified zanamivir, and that the source of this enhanced potency derives primarily from its additional ability to recruit the immune system to eliminate the infection.

A major problem with neuraminidase inhibitors has been their inability to treat infections when administered late in the infection process. In order to investigate whether zan-DNP might prove more effective in treating advanced infections, mice inoculated with a lethal dose of influenza A(H1N1) virus were treated IN with 0.5 µmol/kg zan-DNP (0.7 mg/kg) as described above, only treatment was not initiated until 48, 72, or 96 h after virus inoculation. As shown in Fig. 4b, virus-induced weight loss was minimal in the cohort that was first treated 48 h post-infection, but somewhat more prominent in the 72 h and 96 h cohorts. Moreover, 100% of the animals in the 48 h cohort and 80% of the animals in the 72 h cohort survived the infection. Furthermore, when the dose of zan-DNP was increased from 0.5 to 1.5 µmol/kg, a similar therapeutic effect was observed (Supplementary Fig. 7b). In contrast, identical studies performed on zanamivir-treated mice revealed an unimpeded weight loss in the 48 h cohort, with all animals either dying from infection or requiring euthanasia (due to weight loss) in this cohort. Similar to Fig. 4a, these data also demonstrate that zan-DNP is more effective in treating an influenza A(H1N1) virus infection than free zanamivir.

Because use of an inhaler 2x/day for 5 days might prove inconvenient for some patients, the question next arose whether the frequency or duration of treatment with zan-DNP might be reduced as a means of improving compliance. To explore this issue, a dose frequency study was performed in which zan-DNP was either administered IN 1x/day for 5 days, every other day for 6 days (i.e., 3x) or only once at 24 h post-infection. As seen in Supplementary Fig. 7c, all three dosing regimens, including the single dose at 24 h, eliminated any virus-induced death and prevented infection-induced weight loss. Moreover, as shown in Fig. 4c, the single dose of zan-DNP was also able to prevent any deaths in multiple strains of both influenza A and B virus-infected mice, even though all of the zanamivir alone treated mice died from the infection. Importantly, this remarkable difference in potency could also be demonstrated by monitoring virus titers in the lungs of the infected mice as a function of time following infection (Supplementary Fig. 8). Collectively, these studies demonstrate that a single treatment with zan-DNP should prove sufficient to treat an influenza virus infected patient.

**Parenteral administration of zan-DNP is equally potent.** Unfortunately, some patients cannot use an inhaled neuraminidase inhibitor, either because congestion in their lungs obstructs inhibitor access to the infected alveolar epithelial cells or because the inhaled inhibitor is not well-tolerated[27]. For these patients, an intravenous (IV) formulation of zanamivir has recently proven to be highly beneficial[28]. To determine whether a

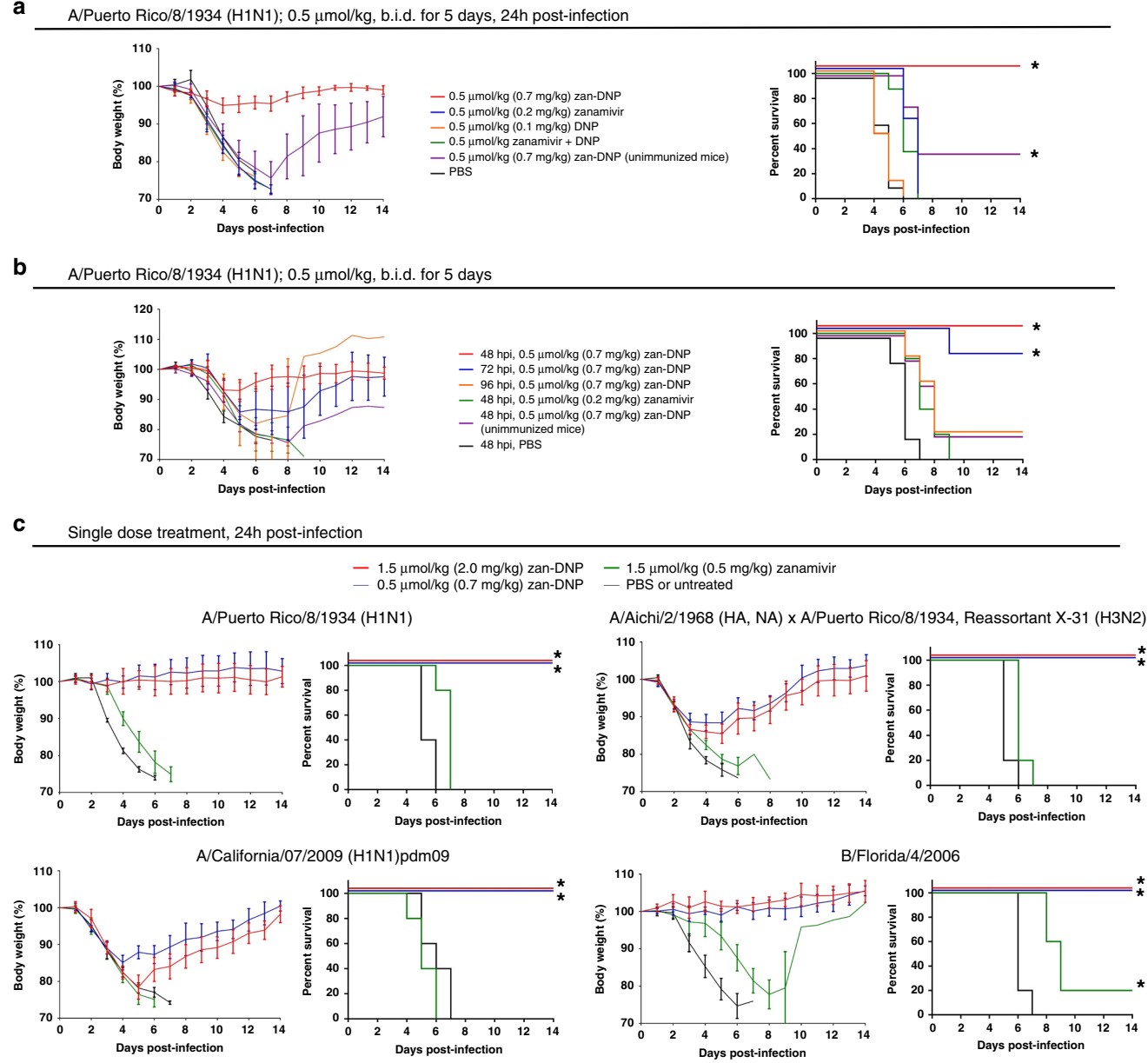

**Fig. 4 Therapeutic efficacy of zan-DNP in protecting mice from lethal influenza virus infections.** DNP-KLH immunized BALB/c mice were challenged with 100x MLD$_{50}$ of the different strains of influenza virus indicated above. Mice were treated with the indicated test articles by intranasal administration (IN) using the dosage regimens indicated above each panel. Mice were counted as dead when they lost either 25% of their initial weight or became moribund. Body weight curves (left panels) and survival curves (right panels) for each treatment are shown. (**a**) compares the efficacy of zan-DNP with its component parts (5 mice/group for zan-DNP and zanamivir treatment groups, 8 mice/group for the rest of the groups). (**b**) examines the effect of different delays between virus inoculation and zanamivir or zan-DNP administration on weight loss and animal survival (5 mice/group). (**c**) investigates the potency of a single dose of either zanamivir or zan-DNP on the same parameters in mice infected with viruses indicated above (5 mice/group). Statistical differences between PBS and drug treatment groups were determined by two-sided log-rank test (*$P < 0.005$, see exact P values in Supplementary Table 1). Body weight change (%) are presented as mean values ± SD.

parenterally administered (i.e., tail vein injected) zanamivir-drug conjugate might similarly gain access to virus-infected lung cells, we first prepared a zanamivir-chelate conjugate that binds $^{99m}$Tc with high affinity by replacing the DNP moiety of zan-DNP with a chelator of $^{99m}$Tc (Fig. 5a, also see Supplementary Fig. 9 for the radiochemical purity of zan-$^{99m}$Tc). Analysis of zan-$^{99m}$Tc binding to influenza virus-infected MDCK cells revealed that the conjugate associates with the influenza virus-infected cells with an apparent $K_d = 15.1$ nM (Fig. 5b). Evidence that this binding is specific for influenza-expressed neuraminidase is provided by the observation that this binding is quantitatively inhibited upon

competition with 100-fold excess of free zanamivir. More importantly, demonstration that IV administered zan-$^{99m}$Tc localizes specifically to virus/virus-infected cells in lungs of influenza virus inoculated mice is supplied by biodistribution data in which influenza virus-infected or uninfected mice were injected IV (tail vein) with zan-$^{99m}$Tc and then analyzed 4 h later for radioactivity in all major tissues. As shown in Fig. 5c, the highest percent injected dose per gram tissue (% ID/g) was found in the lungs (red bar), which constitutes the primary organ in which influenza virus locates. More importantly, this uptake is shown to be totally viral neuraminidase-specific by its absence in mice co-

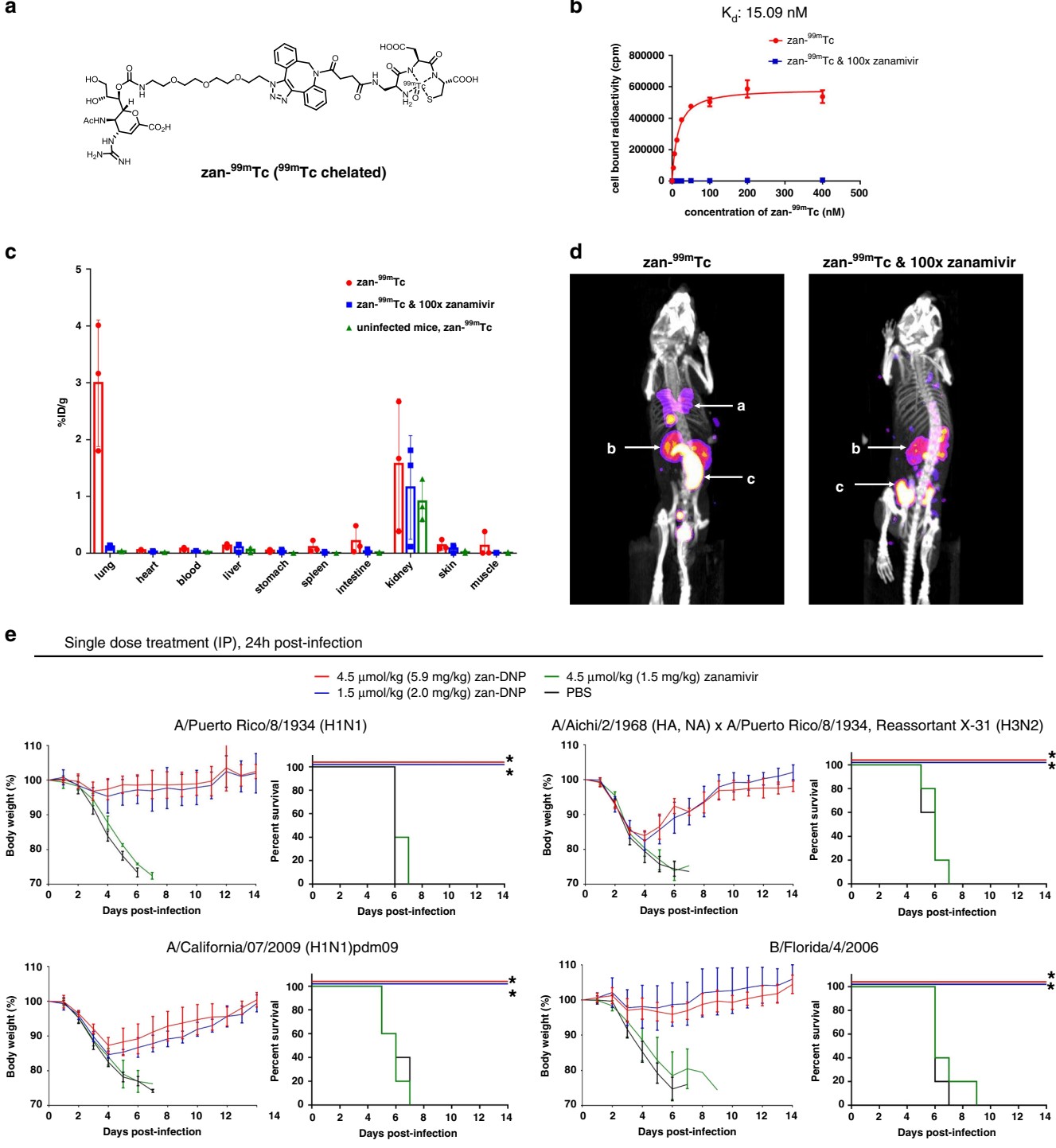

**Fig. 5 Examination of the properties of parenteral administration of zan-DNP. a** Structure of zanamivir-⁹⁹ᵐTc conjugate (zan-⁹⁹ᵐTc). **b** Binding of zanamivir-⁹⁹ᵐTc to N1 neuraminidase (expressed on A/Puerto Rico/8/1934 (H1N1) virus-infected MDCK cells) in the absence (red curve) and presence (blue curve) of 100-fold excess zanamivir (competitor) (n = 3). **c** Biodistribution of intravenously injected (IV) zan-⁹⁹ᵐTc in A/Puerto Rico/8/1934 (H1N1) virus-infected mice (3 mice/group). **d** SPECT/CT images showing the locations of intravenously injected zan-⁹⁹ᵐTc in A/Puerto Rico/8/1934 (H1N1) virus-infected mice (left panel) or virus-infected mice which concurrently received 100-fold excess free zanamivir (right panel) (3 mice/group). a: lung; b: kidney; c: bowels. **e** Weight loss and survival curves of virus-infected mice treated with zan-DNP or zanamivir. DNP-KLH immunized BALB/c mice (5 mice/group) were challenged with 100x MLD₅₀ of the influenza virus strains indicated above each panel. Mice were treated with a single dose of zan-DNP by intraperitoneal administration (IP) 24 h post-infection. Mice were counted as dead when they lost either 25% of their initial weight or became moribund. Statistical differences between PBS and drug treatment groups were determined by two-sided log-rank test (*P < 0.005, see exact P values in Supplementary Table 1). Data are presented as mean values ± SD.

injected IV with 100-fold excess free zanamivir (blue bar). The fact that lungs from uninfected mice injected IV with the same dose of zan-$^{99m}$Tc exhibit no uptake (green bar) further confirms this specificity. While some radioactivity is noted in the kidney, this zan-$^{99m}$Tc is almost certainly undergoing excretion, since the uptake is not significantly reduced by 100-fold excess free zanamivir nor absent from kidneys in uninfected mice. Importantly, all other tissues/organs displayed only background retention of zan-$^{99m}$Tc, confirming that tissue accumulation of zan-$^{99m}$Tc is specifically viral neuraminidase mediated. Further documentation of this virus specificity can be seen in the SPECT-CT images of Fig. 5d, where a strong zan-$^{99m}$Tc signal is observed in the lungs of virus-infected mice (left panel) but absent from infected mice co-injected with excess zanamivir. Interestingly, there is no detectable zan-$^{99m}$Tc signal observed in the nose and trachea. This may be due to the fact that the amount of virus and virus-infected cells in these sites is much lower than that in the lung. Taken together, these studies thus confirm that parenteral administration of zanamivir-linked drug constitutes a feasible route for delivery of antiviral drugs to infection sites in the lungs.

Next, to explore whether mice inoculated with the usual lethal dose of influenza virus might be rescued with a single parenteral dose of zan-DNP, mice that were lethally-infected with either type A [PR8 (H1N1), X31 (H3N2), or A/California/07/2009 (H1N1)pdm09] or type B [B/Florida/4/2006] influenza viruses were injected intraperitoneally (IP) 24 h post-infection with 1.5 or 4.5 μmol/kg zan-DNP or 4.5 μmol/kg zanamivir and then monitored for weight loss and overall survival (Fig. 5e, also see Supplementary Fig. 10 for dosing ranging and comparison studies). As revealed in the figure, all mice treated with unmodified zanamivir died within 7 days of virus inoculation, whereas mice treated with even the lowest concentration of zan-DNP all survived. Although both zan-DNP and zanamivir treated mice experienced a similar weight loss over the initial 3 days of infection, all of the zan-DNP treated mice recovered their normal body weights by 10 days post-infection whereas the zanamivir treated mice continued their uninterrupted weight loss until death. The fact that a high level of protection was observed when neuraminidases from both influenza A and B viruses were targeted confirms the broad-spectrum of activity that characterizes this therapy.

**Analysis of efficacy of exogenous anti-DNP antibodies in vivo.** Finally, because some influenza virus patients may have naturally occurring anti-DNP antibody titers that are too low for therapeutic efficacy (see endogenous titer analyses among healthy volunteers in Supplementary Fig. 3), we elected to evaluate whether co-injection of exogenous anti-DNP antibodies might successfully substitute for endogenous anti-DNP antibodies in mediating an effective anti-influenza immune response. For this purpose, non-immunized mice were inoculated with the usual lethal dose of influenza A(H1N1) virus (i.e., 100× MLD$_{50}$) and then treated 24 h later with 1.5 μmol/kg zan-DNP (2.0 mg/kg) plus different concentrations of exogenous anti-DNP antibodies ranging from 1 to 10 mg/kg. As shown in Fig. 6a, all three doses of exogenous anti-DNP antibodies mediated total recovery from weight loss and complete responses in virus-infected mice, whereas (i) substitution of zanamivir for zan-DNP, (ii) omission of anti-DNP antibody, or (iii) treatment solely with PBS yielded no recoveries from weight loss and incomplete responses to lethal infections. Furthermore, to confirm the generality of the efficacy of anti-DNP antibody supplementation, we performed similar studies on another commonly used laboratory strain (X31 H3N2) plus two additional circulating strains (A/California/07/2009 (H1N1)pdm09 and B/Florida/4/2006). As anticipated, zan-DNP,

but not unmodified zanamivir, was found to yield complete responses in all virus-infected mice in the presence of exogenous anti-DNP antibodies (Fig. 6b). These data demonstrate that injection of exogenous antibodies can be exploited to compensate for any insufficiency in endogenous antibodies in mediating a zan-DNP therapy of an influenza virus infection.

## Discussion

Because multidrug combination therapies generally outperform monotherapies, it was not surprising that a combination of neuraminidase inhibitor with an immunogenic hapten would improve on the potency of the neuraminidase inhibitor. However, with data documenting that individuals receiving an influenza vaccine can still contract infections from viral strains contained in the vaccine[29], together with studies showing that neuraminidase inhibitors can only mitigate an influenza infection when administered early during the infection process[6], the potent eradication by zan-DNP of advanced infections involving very high viral loads was not anticipated. Thus, the ability of zan-DNP to yield complete responses in mice infected with 100 times the MLD$_{50}$ of influenza virus up to 3 days after inoculation was not predicted from data on vaccinated individuals receiving the traditional nontargeted zanamivir. Perhaps the fact that each virus-infected cell is concurrently exposed to two orthogonal therapies; i.e., one that blocks release of budding virus and a second that promotes immune-mediated destruction can account for the heightened potency. Alternatively, the fact that most current vaccines are targeted to viral hemagglutinins[30,31], whereas the immune component of zan-DNP targets viral neuraminidases may somehow explain the unexpected potency.

In addition to remarkable potency, several other features of zan-DNP render attractive for further development. First, unlike the traditional flu vaccines, the immune component of the zan-DNP therapy does not require an ability to predict which viral strains will emerge during the forthcoming flu season[5]. Because zanamivir is believed to treat infections caused by all known subtypes/lineages of influenza A and B viruses[19], the identity of the strains most likely to propagate should become less important. Second, the route of zan-DNP administration may afford the physician additional treatment options not available with zanamivir. Thus, mice receiving zan-DNP either intranasally or intraperitoneally were found to respond to drug roughly equivalently. If airway congestion or other physical limitations were to be found to obstruct administration of zan-DNP intranasally, the physician would have the option to administer the same dose of drug intraperitoneally, subcutaneously or intravenously. Third, unlike other neuraminidase inhibitors, binding of zan-DNP to released viral particles should contribute to free virus elimination, whereas zanamivir binding is not thought to recruit immune involvement. For example, antibody/complement opsonized viral particles would still be expected to undergo phagocytosis by Fc receptor expressing cells, whereas zanamivir bound viruses would not. Finally, as noted above, zan-DNP (unlike other neuraminidase inhibitors) appears to lose potency only late in the infection/propagation process (see Fig. 4b), whereas zanamivir has been found to decline in potency rapidly after symptoms first appear[6].

It has not escaped our attention that other targeting ligands and immunogenic haptens could have easily been used instead of zanamivir and/or DNP to create related ligand-hapten immunotherapies. Thus, influenza hemagglutinins are similarly exported to the surface of virus-infected cells and hemagglutinin ligands are similarly available to deliver haptens to these viral receptors[32,33]. Moreover, influenza M2 proteins are also trafficked to infected cell surfaces and should be similarly accessible for

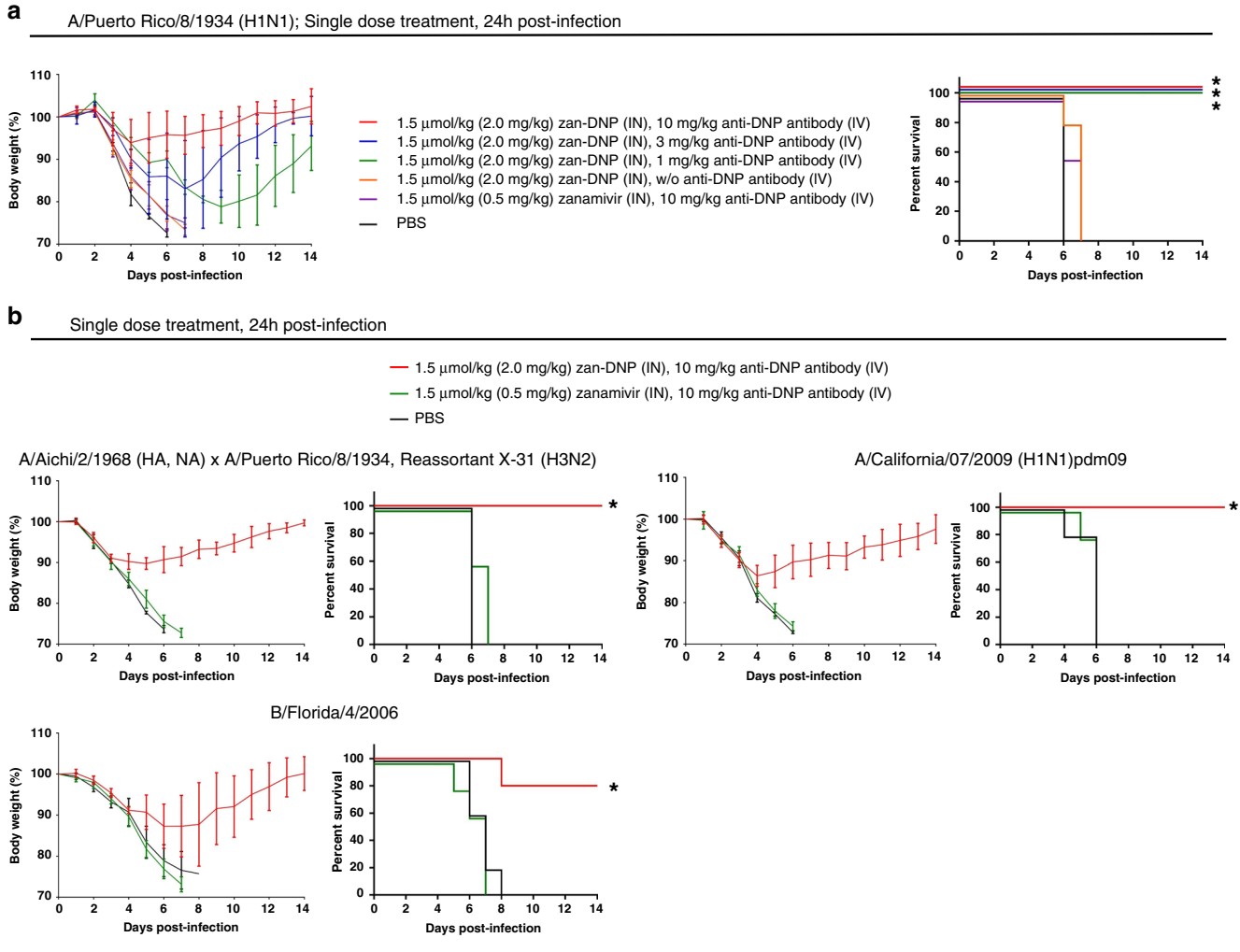

**Fig. 6 Therapeutic efficacy of co-administration of zan-DNP plus exogenous anti-DNP antibodies in protecting nonimmunized mice from a lethal influenza virus infection. a** Nonimmunized BALB/c mice were challenged with 100x $MLD_{50}$ of influenza virus A/Puerto Rico/8/1934 (H1N1) prior to treatment with a single dose of 1.5 μmol/kg zan-DNP (intranasal administration, IN) plus exogenous anti-DNP antibodies (1–10 mg/kg, intravenous administration, IV) 24 h post-infection (3 mice/group for 3 mg/kg and 1 mg/kg anti-DNP antibody treatment groups, 5 mice/group for the rest of the groups). Control groups include (i) substitution of zanamivir for zan-DNP, (ii) omission of anti-DNP antibodies, and (iii) treatment with PBS. **b** Nonimmunized BALB/c mice (5 mice/group) were challenged with the indicated strains of influenza virus prior to treatment with a single dose of 1.5 μmol/kg zan-DNP or zanamivir (IN) plus 10 mg/kg exogenous anti-DNP antibodies (IV) 24 h post-infection. Mice were counted as dead when they lost either 25% of their initial weight or became moribund. Statistical differences between PBS and drug treatment groups were determined by two-sided log-rank test (*$P < 0.005$, see exact $P$ values in Supplementary Table 1). Body weight change (%) are presented as mean values ± SD.

ligand-hapten binding[14]. Since additional haptens are known against which endogenous anti-hapten antibodies pre-exist in human sera (e.g., anti-rhamnose, anti-alpha-galactosyl, anti-tetanus, etc.)[18], opportunities to develop totally orthogonal ligand-hapten conjugates that could similarly promote eradication of virus-infected cells seem abundant. Thus, if strains of virus were ever to emerge that exhibited reduced affinity for zanamivir, additional ligand-hapten conjugates could be developed to augment the zan-DNP therapy.

Because many life forms including humans, other mammals and even bacteria express an isoform of neuraminidase, the question naturally arises whether off-target recognition of an endogenous neuraminidase might lead to unwanted toxicity. While an absolute answer to this concern must await human clinical trials, it should be noted that most bacterial neuraminidases bind zanamivir with $IC_{50}$ values ranging from 0.1 to 5 mM[34]; i.e., one million times weaker than influenza virus neuraminidase ($IC_{50}$ ~0.5–3 nM), and the four isoforms of human neuraminidase enzymes bind zanamivir at least a few thousand

times weaker than influenza neuraminidase[35]. When considered with the observation that little if any non-specific uptake of zanamivir-$^{99m}$Tc conjugate was seen in mice not inoculated with influenza virus (Fig. 5c, green bars), one can conclude that dose-limiting target toxicity to healthy tissues is not very likely.

The use of a low molecular weight homing ligand to target an attached drug specifically to a diseased cell type has received growing interest in recent years. Thus, ligand-targeted drugs have been developed for treatment of multiples cancers[23], autoimmune diseases[36,37], fibrotic diseases, and tissue traumas[38], with improved potencies and decreased off-target toxicities reported in most cases. Although direct comparisons have not been made, the specificity of zan-DNP for its disease-specific receptor appears to be much greater than the specificity of any previous ligand-targeted drug for its cognate receptor (Fig. 5c), i.e., with competable uptake being restricted almost exclusively to the lungs of infected mice. Although the previously targeted receptors (e.g., folate receptor, prostate specific membrane antigen, carbonic anhydrase IX, EGF receptor, etc.) all exhibit expression in a subset

of healthy cells[23], zan-DNP should have no healthy cell counterpart, perhaps accounting for its virus-infected cell specificity.

Finally, it should be emphasized that the ligand-hapten targeting strategy for killing pathologic cells has already been explored for treatment of many other diseases. Thus, the first ligand-hapten immunotherapy was applied to eradication of folate receptor overexpressing cancers in both murine tumor models[39] and human patients with renal cancer[40]. Although folate replaced zanamivir and fluorescein replaced DNP in these studies, complete tumor eradications were seen in animal studies and partial responses with minimal toxicities were observed in the phase 1 human clinical trials. Perhaps encouraged by these data, related hapten-targeted immunotherapies were subsequently developed for treatment of melanomas[41] and prostate cancers[42], with very promising data reported in animal models in both cases. Hapten-targeted immunotherapies have also been explored for suppression of activated macrophages in murine models of inflammatory and autoimmune diseases[36,37], with animal data again demonstrating impressive efficacy in all cases. More recently, related ligand-targeted hapten therapies have been developed to kill HIV-, bacteria- and fungus-infected cells, with analogous promising data again reported in all cases[18,43,44]. Taken together, these congruent data from multiple labs suggest that decoration of a pathogenic cell with a strongly immunogenic hapten can mediate eradication of the pathogenic cell in a variety of diseases. And although any cell decorated with DNP hapten will likely be destroyed by the immune system, the data in Fig. 5c showing receptor expression limited to infected tissues specifically argue that off-target toxicities should be minimal.

## Methods

**Compound Synthesis**. The detailed syntheses of zanamivir-related influenza neuraminidase targeting ligand, zan-DNP, zan-rhodamine, and zan-$^{99m}$Tc are shown in Supplementary Information.

**Cell lines/viruses/mice**. Madin-Darby canine kidney (MDCK) cells were obtained from American Type Culture Collection (ATCC). The cell growth medium was prepared according to the *Manual for the laboratory diagnosis and virological surveillance of influenza*[45]: Dulbecco's Modified Eagle Medium (DMEM) (high glucose, Invitrogen—cat. no. 11965-092), 10% heat inactivated fetal bovine serum (HIFBS), 2% bovine albumin fraction v (BSA) (7.5%), 2% HEPES (1 M), and 1% penicillin-streptomycin. Human embryonic kidney 293 (HEK 293) cells were obtained from ATCC and grown in DMEM medium containing 10% HIFBS, 1% HEPES (1 M), and 1% penicillin-streptomycin. All cells were grown at 37 °C in a humidified atmosphere (5% $CO_2$: 95% air).

Normal human bronchial epithelial (NHBE) cells were purchased from Lonza and cultured in bronchial epithelial cell growth medium (BEGM, Lonza). To establish the cell culture at air–liquid-interface (ALI), NHBE cells at passage two were seeded on 12 mm transwell inserts in 12-well plates (Corning) and grown submerged in BEGM. When cells reached completely confluency, the medium in apical chambers was removed and the medium in basal chambers was replaced with HBTEC Air-Liquid Interface Differentiation Medium (Lifeline Cell Technology). The cells were maintained at ALI for at least 21 days to ensure full differentiation. The basal medium was changed every 2 days and the cells were washed with HBSS weekly to remove mucus.

Influenza virus A/Puerto Rico/8/1934 (H1N1) (Catalog No. NR-348), A/Aichi/2/1968 (HA, NA) × A/Puerto Rico/8/1934 (H3N2) reassortant X-31 (Catalog No. NR-3483), A/Wisconsin/629-D00015/2009 (H1N1)pdm09 (Catalog No. NR-19806), A/Netherlands/22/2003 (H3N2) (Catalog No. NR-49236), B/Florida/4/2006 (Yamagata Lineage) (Catalog No. NR-41795), B/Brisbane/60/2008 (Victoria Lineage) (Catalog No. NR-42006) and A/California/07/2009 (H1N1)pdm09 (non-mouse-adapted virus, Catalog No. NR-13663) were obtained from BEI Resources, NIAID, NIH. Influenza virus A/Puerto Rico/8/1934 (H1N1) (tissue culture adapted, Catalog No. VR-1469™) and A/Aichi/2/1968 (H3N2) (Catalog No. VR-1680™) were purchased from ATCC.

Female BALB/c mice (3–4 week old: up to 12 g; 6–9 week old: 16 to 21 g) were purchased from Envigo (location: Indianapolis, IN, United States). All animal procedures were approved by Purdue Animal Care and Use Committee (PACUC).

**$TCID_{50}$ endpoint dilution assay**. Infectious influenza virus titers were determined by a 50% tissue culture infective dose ($TCID_{50}$) endpoint dilution assay[46]. For this purpose, serial 10-fold dilutions of influenza virus were mixed with equal volumes of MDCK cells at $2 \times 10^5$ cells/mL (all the virus and cell solutions were prepared

using virus growth medium: high glucose DMEM containing 2% BSA, 2% HEPES (1 M), 1% penicillin-streptomycin and 2 μg/ml TPCK-trypsin). The mixtures of cells and virus were then added to 96-well plates at 100 μl/well. After incubation at 37 °C under 5% $CO_2$ for 48 h, cell viability was measured by CellTiter 96® AQueous Non-Radioactive Cell Proliferation Assay (Promega). $TCID_{50}$ was calculated using the Reed–Muench method[47].

**Preparation of influenza virus infected-MDCK cells**. Infections of MDCK cells with influenza virus were performed using a standard method[45]. Low-passage MDCK cells were seeded in 24-well plates or confocal microscope plates. When cells reached confluence, cell growth medium was aspirated, and the cells were washed 2x with phosphate buffered saline (PBS). Cells were then inoculated with influenza virus at 100 $TCID_{50}$ (100 μl/well) and incubated to absorb virus at 37 °C for 1 h. During the incubation period, the plates were tapped every 15 min to redistribute the inoculum. At the end of the inoculation, 500 μl virus growth medium (high glucose DMEM containing 2% BSA, 2% HEPES (1 M), 1% penicillin-streptomycin and 2 μg/ml TPCK-trypsin) was added to each well and the plates were incubated at 37 °C under 5% $CO_2$ to allow the virus-infected MDCK cells to express influenza neuraminidase.

**Saturation binding assay**. The binding affinity of zan-rhodamine for neuraminidases (expressed on influenza virus-infected cells) was measured by saturation binding assay. Influenza virus-infected MDCK cells in 24-well plates were prepared as described above, and on the day of the experiment spent medium was aspirated and virus-infected cells were incubated at 37 °C for 1 h with 0.5 mL of fresh medium containing increasing concentrations of zan-rhodamine in the presence or absence of 100-fold excess of unmodified zanamivir. Virus-infected cells were then rinsed with fresh medium (2 × 0.5 mL) to remove unbound zan-rhodamine and dissolved in 0.5 mL of 1% aqueous sodium dodecyl sulfate (SDS). In total 200 μl of SDS solution from each well was transferred to 96-well black walled plates (Corning) and cell-associated fluorescence was measured by Synergy Neo2 HTS Multi-Mode Microplate Reader (Biotek) using an excitation of 560 nm and emission of 620 nm. The dissociation constant ($K_d$) was calculated by plotting the cell bound fluorescence intensity versus the concentrations of zan-rhodamine using GraphPad Prism 7 (Saturation binding equations, One site—Total and nonspecific binding).

For the assays performed with NHBE cells grown at ALI, the cells were washed with PBS to remove the mucus and then apically inoculated with influenza virus (MOI of 1, measured in MDCK cells) diluted in 100 μl BEGM without TPCK-trypsin. After inoculation at 37 °C for 1 h, the inoculum was aspirated, and the cells were incubated at 37 °C for 36 h before use. On the day of the experiment, the basal medium was aspirated and virus-infected cells were incubated at 37 °C for 1 h with 0.3 mL of fresh medium containing increasing concentrations of zan-rhodamine in the presence or absence of 100-fold excess of unmodified zanamivir. Virus-infected cells were then rinsed with fresh medium (2 × 0.3 mL) to remove unbound zan-rhodamine and dissolved in 0.3 mL of 1% aqueous sodium dodecyl sulfate (SDS). In total 200 μl of SDS solution from each transwell insert was transferred to 96-well black walled plates (Corning) and cell-associated fluorescence was measured, as mentioned above.

**Competitive binding assay**. The binding affinity of zanamivir or zan-DNP for viral neuraminidases was measured using a competitive binding assay. Influenza virus-infected MDCK cells in 24-well plates were prepared, as described above. On the day of each experiment, spent medium was aspirated and virus-infected cells were incubated for 1 h at 4 °C with 0.5 mL of fresh medium containing 15 nM zan-rhodamine plus increasing concentrations of either zanamivir or zan-DNP. Virus-infected cells were then rinsed with fresh medium (2 × 0.5 mL) to remove unbound zan-rhodamine and dissolved in 0.5 mL of 1% aqueous sodium dodecyl sulfate (SDS). Cell-associated fluorescence was measured using the plate reader as described above. The dissociation constant ($K_d$) was calculated by plotting the cell bound fluorescence intensity versus the log of the concentration of unlabeled compound using GraphPad Prism 7 (competition Binding equations, One site – Fit $K_i$).

**In vitro virus inhibition assay**. The in vitro antiviral activity of the compounds was evaluated in MDCK cells and expressed as the concentration of compound that protects 50% of cells from virus-induced death ($EC_{50}$). MDCK cells were harvested and plated in 96-well plates at $2 \times 10^5$ cells/mL (50 μl). Serial 3-fold dilutions of each compound were then mixed with equal volumes of influenza virus at 100 $TCID_{50}$. Following ~10 min incubation, the mixtures of compound plus virus (50 μl) were added to MDCK cells and the suspension was incubated at 37 °C for 48 h under 5% $CO_2$. Cell viability was then measured using CellTiter 96® AQueous Non-Radioactive Cell Proliferation Assay (Promega). $EC_{50}$ values were measured by plotting the absorbance at 450 nm (indicating the viability of the cells) versus the log of each compound's concentration using GraphPad Prism 7.

**Confocal microscopy analysis of anti-DNP antibody recruitment**. Antibody recruitment by cell surface bound zan-DNP was analyzed by culturing influenza virus-infected MDCK cells in confocal microscope plates as described above and on the day of each experiment removing spent medium and incubating the virus-infected cells for 30 min at 37 °C with 0.5 mL of fresh medium containing 50 nM

zan-DNP in the presence or absence of 5 μM zanamivir. Virus-infected cells were then washed with fresh medium (2 × 0.5 mL) and incubated for 30 min at 37 °C with 0.5 mL of 20 μg/mL anti-DNP IgG-biotin conjugate (Invitrogen, cat. no. A-6435). Virus-infected cells were then incubated at 37 °C for 30 min with 0.5 mL of 5 μg/mL streptavidin-phycoerythrin (PE) conjugate (BD Biosciences). After the final wash (2 × 0.5 mL), the virus-infected cells were fixed with 4% paraformaldehyde in PBS and cell nuclei were stained with TO-PRO®-3 Iodide (Invitrogen). Confocal images were then acquired by confocal microscopy (FV 1000, Olympus).

For the assays performed with NHBE cells grown at ALI, the cells were washed with PBS to remove the mucus and then apically inoculated with influenza virus (MOI of 1, measured in MDCK cells) diluted in 100 μl BEGM without TPCK-trypsin. After inoculation at 37 °C for 1 h, the inoculum was aspirated, and the cells were incubated at 37 °C for 36 h before use. On the day of the experiment, the basal medium was removed, and the virus-infected cells were incubated for 30 min at 37 °C with 0.3 mL of fresh medium containing 50 nM zan-DNP in the presence or absence of 5 μM zanamivir. Virus-infected cells were then washed with fresh medium (2×0.3 mL) and incubated for 30 min at 37 °C with 0.3 mL of 10 μg/mL human anti-DNP IgG1 (ACROBiosystems, cat. no. DNP-M2). Virus-infected cells were then incubated at 37 °C for 30 min with 0.3 mL of 5 μg/mL goat anti-human F (ab′)$_2$-PE (Abcam, cat. no. ab98596). To identify virus-infected cells, the cells were concurrently stained with 0.3 mL of 10 μg/mL rabbit anti-hemagglutinin antibody (Sino Biological Inc, cat. no. 86001-RM01) followed by 0.3 mL of 10 μg/mL goat anti-rabbit IgG-AF647 (Invitrogen, cat. no. A-21244). After the final wash (2 × 0.3 mL), the transwell inserts were placed on glass slides and confocal images were acquired by confocal microscopy (FV 1000, Olympus).

**Preparation of neuraminidase-expressing HEK293 cell line**. The coding sequence for neuraminidase from influenza virus A/Puerto Rico/8/1934 (H1N1) was cloned into lentiviral transfer vector pWPI (addgene). The transfer vector was then co-transduced into HEK293TN cells using the Ready-to-Use Packaging Vector mixture (Cellecta) plus Lipofectamine 3000 Transfection Reagent (Invitrogen). Culture supernatants were harvested and used to transduce HEK293 cells. After 48 h of transduction, cells were sorted by flow cytometry for GFP positive cells and further subcloned. Subcloned line was used as target cells in CDC and ADCC assays described below.

**Complement dependent cytotoxicity (CDC) assay**. Analysis of virus-infected cell killing by CDC was conducted by an CDC assay[44]. N1 transduced HEK 293 cells and nontransduced HEK 293 cells were harvested and plated in triplicate (25 μl, $4 \times 10^5$ cells/mL) in 96-well black walled plates (Corning) and then treated with serial 5-fold dilutions of zan-DNP in the presence or absence of 100-fold excess of zanamivir. After incubating at rt for 30 min, 50 μl of a solution containing 40 μg/mL rabbit anti-DNP IgG (Invitrogen, cat. no. A-6430) and 20% (v/v) rabbit complement serum (Sigma-Aldrich) was added to each well and the plates were incubated for 4 h at 37 °C under 5% $CO_2$. One hundred percent cell death was achieved by adding 5% $H_2O_2$ to the HEK 293 cells and cell viability was measured using CellTiter-Glo® Luminescent Cell Viability Assay (Promega). The percent zan-DNP mediated CDC was calculated as: $100 \times$ (luminescence $_{\text{no zan-DNP}}$–luminescence $_{\text{expt}}$)/(luminescence $_{\text{no zan-DNP}}$–luminescence $_{\text{maxkilling}}$).

**Antibody-dependent cellular cytotoxicity (ADCC) assay**. Analysis of ADCC was performed using an ADCC kit (Promega, cat. no. G7010) plus N1-transduced and nontransduced HEK 293 cells. For this purpose, cells were plated in triplicate (25 μl, $4 \times 10^5$ cells/mL) in 96-well black walled plates (Corning) and then treated with serial 5-fold dilutions of zan-DNP in the presence or absence of 100-fold excess zanamivir. After incubating at rt for 30 min, 25 μl of 33.3 μg/mL human anti-DNP IgG1 (ACROBiosystems) were added to each well and the plates were incubated at rt for 30 min. Finally, ADCC effector cells were added at 75,000 cells/well and incubated for 6 h at 37 °C under 5% $CO_2$. The amount of firefly luciferase produced by ADCC effector cells was then quantified using Bio-GloTM Luciferase Assay Reagent (included in the kit). Luminescence was measure using Synergy Neo2 HTS Multi-Mode Microplate Reader (Biotek). % zan-DNP mediated ADCC was calculated as: luminescence $_{\text{expt}}$–luminescence $_{\text{no zan-DNP}}$.

**Mouse immunization**. Induction of anti-DNP antibodies in mice was performed according to a published method[42]. DNP-hemocyanin conjugate (DNP-KLH, MilliporeSigma) was dissolved in autoclaved water at 1 mg/ml and emulsified with an equal volume of complete Freund's Adjuvant (CFA, Sigma-Aldrich) or incomplete Freund's Adjuvant (IFA, Sigma-Aldrich). The emulsification was conducted through a stepwise addition method[48]. In total 3–4 week old BALB/c mice were immunized subcutaneously at the base of the tail first with 200 μl of CFA emulsified DNP-KLH solution and then 2 weeks later with 200 μl of IFA emulsified DNP-KLH solution. Mouse serum samples were collected 2 weeks later to test for the presence of anti-DNP antibodies.

**ELISA for detection of anti-DNP antibodies**. ELISA 96-well plates (Fisher Scientific) were coated at 4 °C overnight with 100 μl of 5 μg/ml BSA-DNP (Invitrogen) dissolved in coating buffer (1.0 L deionized water containing 8.4 g $NaHCO_3$ and 3.56 g $Na_2CO_3$, pH: 9.5). Plates were washed 3x with ELISA wash buffer

(BioLegend) and incubated with blocking buffer (10% FBS in PBS) at rt for 1 h to block nonspecific binding. After washing 4x with ELISA wash buffer, wells were treated with 100 μl of serial 10-fold dilutions of human/mouse serum in blocking buffer and incubated at rt for 2 h. Plates were then washed 4x with ELISA wash buffer and incubated for 2 h at rt with 100 μl of anti-human IgG-HRP, anti-human IgM-HRP, anti-mouse IgG-HRP or anti-mouse IgM HRP (1:5000 dilution in blocking buffer, Invitrogen). After washing 4x with ELISA wash buffer, plates were treated with 100 μl/well freshly prepared TMB substrate solution (BioLegend) and allowed to react for 5 min before the enzymatic reactions were terminated by adding 100 μl 1 N HCl. The optical density was then read at 450 nm (O.D.) using Synergy Neo2 HTS Multi-Mode Microplate Reader (Biotek), and the results were plotted as average absorbance values at 450 nm versus log serum dilution factors.

**Virus challenge and mouse therapy studies**. Determination of the viral titer required to kill 50% of inoculated mice ($MLD_{50}$) was achieved by anesthetizing 6–9 week old BALB/c mice (5 mice/group) with isoflurane and inoculating the mice intranasally (IN) with 50 μl of serial 10-fold dilutions of influenza virus. Mice were weighed and monitored daily for 14 days post-infection and counted as dead when they either lost 25% of their body weight or were diagnosed as moribund. $MLD_{50}$ titers were then calculated using the Reed–Muench method[47].

For evaluation of drug efficacy in mice immunized with DNP-KLH, BALB/c mice were challenged IN with 100x $MLD_{50}$ of influenza virus. During the following two weeks, mice (5 mice/group) were then treated with different concentrations of test articles (zan-DNP, zanamivir, DNP or PBS) according to the indicated dosing regimens. In the studies in which mice were treated IN with test articles, mice were anesthetized with isoflurane before the treatment. Mice were then analyzed as above.

For analysis of therapeutic efficacy in non-immunized mice, mice (5 mice/group) were first challenged IN with 100x $MLD_{50}$ of influenza virus and then administered (24 h post-infection) a single IN dose of drug (zan-DNP, zanamivir, or PBS), mice were anesthetized with isoflurane before the treatment) followed by a single intravenous (IV) dose of rabbit anti-DNP IgG (Invitrogen, cat. no. A-6430). Mice were then analyzed as above.

**Flow cytometry analysis of anti-DNP antibody recruitment**. Around 6–9 week old mice (3 mice/group) were inoculated IN with 100x $MLD_{50}$ of influenza virus A/California/07/2009 (H1N1)pdm09. Three days after infection, mice were administrated with a single dose of 0.5 μmol/kg zan-DNP or zanamivir plus 5 mg/kg rat anti-DNP antibodies (IV) (Invitrogen, cat. no. 04-8300). Twelve hours later, the lungs of virus-infected/uninfected mice were harvested immediately after euthanasia. The right lungs were digested using gentleMACS Octo Dissociator (Miltenyi Biotec) with mouse lung dissociation kit (Miltenyi Biotec, cat. no. 130-095-927). The cell suspension was filtered through 70 μm cell strainer (Miltenyi Biotec) and washed 2x with PBS. The erythrocytes were removed using red blood cell lysis buffer (Biolegend) followed by PBS wash. The resulting cells were resuspended in freezing medium (90% FBS plus 10% DMSO) and stored at –80 °C for future flow cytometry analyses (if the cells are used immediately after digestion, they should be incubated with FBS to block the non-specific bindings of secondary antibodies). After thawing, cells were blocked with TruStain FcX™ antibody (Biolegend) and then labeled with rabbit anti-hemagglutinin antibody (1:100 dilution, Sino Biological Inc, cat. no. 86001-RM01) for 1 h on ice. Cells were washed 3x with PBS and then stained with goat anti-rat IgG-PE (1:500 dilution, Invitrogen, cat. no. A10545) and goat anti-rabbit IgG-AF647 (1:1000 dilution, Invitrogen, cat. no. A-21244) for 20 min on ice. The cells were washed 3x with PBS and analyzed using Attune NxT Flow Cytometer.

**Quantitation of lung viral titers**. DNP-KLH immunized mice were inoculated IN with 100x $MLD_{50}$ of influenza virus A/Puerto Rico/8/1934 (H1N1) as described above and then treated 24 h later with a single IN dose of 1.5 μmol/kg (2.0 mg/kg) zan-DNP, zanamivir, or PBS. Mice were sacrificed by $CO_2$ asphyxiation 3 and 5 days post-infection, and their lungs were harvested and immediately homogenized using gentleMACS Octo Dissociator (Miltenyi Biotec)[49]. Viral titers from the lung homogenates were measured by real time RT-PCR. RNA was extracted from the homogenates using Quick-RNA™ Microprep Kit (ZYMO RESEARCH). cDNA synthesis and reverse transcription were performed according to a standard protocol[50]. The primer/probe sets were synthesized to recognize two highly conserved regions of influenza matrix (M) gene (see sequences in Supplementary Table 2). To construct a standard curve for calculation of viral titers, 10-fold dilutions of influenza virus A/Puerto Rico/8/1934 (H1N1) stock solution with a known viral titer were run in parallel with the lung homogenates.

**Preparation of a zan-$^{99m}$Tc formulation kit**. The following reagents for radiolabeling of zanamivir-chelate precursor (**compound 14**) with $^{99m}$Tc were prepared and purged with argon: sodium gluconate (Sigma-Aldrich) in water (250 mg, 2.5 mL); ethylenediaminetetraacetic acid (EDTA) disodium salt dihydrate (Sigma-Aldrich) in water (30 mg, 3.0 mL); and tin(II) chloride dihydrate (Sigma-Aldrich) in 0.2 N HCl (50 mg, 5 mL). To 2.5 mL of sodium gluconate solution was added 300 μl of EDTA solution and 100 μl of Tin(II) chloride solution while purging with argon. To this solution was added 1.6 mg zanamivir-chelate precursor (1340 nmol; **compound 14**) while bubbling with argon. After adjusting the solution pH to 6.8 using argon purged 1.0 N NaOH and 0.2 N HCl, the volume of the solution was

increased to 10 mL by adding argon-purged water. The final solution was dispensed to formulation vials (1.0 mL/vial) under argon and lyophilized for 48 h. Vials were sealed with rubber stoppers and aluminum seals under argon atmosphere and stored at −80 °C until use.

**Radiolabeling of zanamivir-chelate precursor with $^{99m}$Tc**. When desired, a zan-$^{99m}$Tc formulation vial was warmed to rt and injected through a septum with sodium pertechnetate $^{99m}$Tc solution (1.0 mL, 15 mCi, Cardinal Health) and then swirled gently to dissolve the lyophilized powder. After allowing to stand at rt for 15–20 min, 2 ml of air was injected into the vial to oxidize any excess stannous ion. The solution was then stored at rt and used within 6 h.

**Analysis of zan-$^{99m}$Tc binding to virus-infected MDCK cells**. Binding of zan-$^{99m}$Tc to influenza neuraminidase (expressed on influenza virus A/Puerto Rico/8/1934 (H1N1)-infected cells) was measured by saturation binding assay. Influenza virus-infected MDCK cells in 24-well plates were prepared as described above. On the day of the experiment spent medium was aspirated and virus-infected cells were incubated with 0.5 mL of fresh medium containing increasing concentrations of $^{99m}$Tc labeled zan-$^{99m}$Tc in the presence or absence of 100-fold excess of zanamivir free drug. After incubating for 1 h at 4 °C, virus-infected cells were then rinsed with fresh medium ($3 \times 0.5$ mL) to remove unbound radioligand and dissolved in 0.5 mL of NaOH (0.25 M). Cell lysates from each well were transferred into γ-counter tubes and the cell-bound radioactivity was counted using a γ-counter (Packard, Packard Instrument Company). The dissociation constant ($K_d$) was calculated by plotting the cell bound radioactivity versus the concentrations of zan-$^{99m}$Tc using GraphPad Prism 7 (Saturation binding equations, One site—Total and nonspecific binding).

**Biodistribution and SPECT-CT imaging of zan-$^{99m}$Tc in vivo**. In total 6 to 9 week old mice (3 mice/group) were inoculated IN with 100x $MLD_{50}$ of influenza virus A/Puerto Rico/8/1934 (H1N1). Three days after infection, mice were administrated 100 μl of $^{99m}$Tc labeled zan-$^{99m}$Tc (1.3 nmol, 150 μCi) diluted in PBS via tail vein injection. Mice were sacrificed by $CO_2$ asphyxiation 4 h after injection. Each mouse was dissected, and selected tissues/organs were collected into γ-counter tubes. Radioactivity of weighed tissues/organs were counted and counts per minute (CPM) values were decay corrected. Results are reported as percent injected dose per gram of wet tissue (% ID/g): $100 \times (CPM_{tissue}/CPM_{injected\ dose})/Wt.$ of tissue.

A second batch of 6–9 weeks old mice (3 mice/group) were infected with 50 μl of influenza virus A/Puerto Rico/8/1934 (H1N1) as above, and three days later administered 100 μl of $^{99m}$Tc labeled zan-$^{99m}$Tc (6.5 nmol, 750 μCi) via tail vein injection. Mice were killed by $CO_2$ asphyxiation 4 h after injection and SPECT-CT imaging was performed using micro-SPECT II/CT (MILabs). 3D reconstruction was then conducted using ImageJ software.

**Statistics and reproducibility**. Statistical analyses were performed using Graph-Pad Prism 7 (GraphPad Software, CA). For flow cytometry analysis of anti-DNP antibody recruitment (Fig. 3e) and quantitation of lung viral titers (Supplementary Fig. 8), the statistical difference between two groups were determined using an unpaired two-sided *t*-test with a 95% confidence interval (*$P < 0.05$, **$P < 0.01$, ***$P < 0.001$). For all of the mouse therapy studies, a two-sided log-rank test was applied to compare the survival difference between PBS (or untreated) and drug treatment groups (*$P < 0.005$). For confocal microscopy analysis of anti-DNP antibody recruitment (Fig. 3a, c, Supplementary Fig 1b), the experiments were repeated twice independently with similar results. All other in vitro experiments were performed once (in triplicate). All live mouse experiments were performed once (the number of mice used for each experiment is indicated in figure legends).

**Reporting summary**. Further information on research design is available in the Nature Research Reporting Summary linked to this article.

## Data availability
All data are present in the main text and supplementary information or available from the authors on reasonable request. Source data are provided with this paper.

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

## Acknowledgements

We would like to acknowledge the Purdue Center for Cancer Research, the Purdue Institute for Drug Discovery, the Purdue Flow Cytometry Core, the Purdue Imaging Facility and the Purdue Chemical Genomics Facility for their services and assistance. We thank BEI Resources, NIAID, NIH for providing the influenza viruses used in this study. This research was supported by Endocyte Inc. (Endocyte Grant#: 201194).

## Author contributions

X.L., B.Z., and P.S.L. designed the study. X.L performed the compound synthesis and in vitro and in vivo studies. X.L., B.Z., M.S., and P.S.L. analyzed and interpreted the data. X.L., and P.S.L. wrote the manuscript. B.Z. performed the cell transduction, animal vaccination and part of immunological assays. X.L., H.S.H, and Y.W. cultured the viruses and developed the virus-infected mouse model. L.X. developed the method to prepare the formulation kit and measured the radio-purity of $^{99m}$Tc labeled zanamivir-chelate precursor. F.Z. performed mouse lung digestion and flowcytometry analyses. Y.W. and M.S. helped with compound synthesis. M.S. managed and coordinated the project. P.S.L. conceived and supervised the project.

## Competing interests

Xin Liu and Philip S. Low have applied through Purdue University for a provisional patent that covers the therapy described in this paper. Other authors declare no competing interests.
