## [Peer Review File · Nature Communications]

REVIEWER COMMENTS

Reviewer #1 (Remarks to the Author):

The hapten-targeted immunotherapies is a rapidly developing approach for killing pathologic cells has already been explored for treatment of many diseases, such as melanomas, prostate cancers and other cancers. Hapten-targeted immunotherapies have also been explored for suppression of activated macrophages in murine models of inflammatory and autoimmune diseases, for therapies against HIV-, bacteria- and fungus-infected cells.

In the current study, the authors described the use of zanamivir, a neuraminidase inhibitor of influenza virus, to deliver an attached immunogenic hapten specifically to the surface of virus/virus-infected cells, leading to simultaneous inhibition of virus release and immune-mediated eradication of both free virus and virus-infected cells. This is a novel and very interesting approach in influenza therapy.

MAJOR COMMENTS:

1. Abstract – the composition of a new drug must be clearly described. Abstract requires modifications.
2. The possible administration routes of zanamivir-DNP should be discussed in more details. The neuraminidase inhibitor zanamivir is still not so often prescribed for influenza due to inhalation route of administration.
3. Introduction section requires clarifications about the existing anti-influenza therapies, role of neuraminidase in virus replication cycle, and anti-DNP antibodies.

COMMENTS:

1. Abstract – Seasonal influenza epidemics infect approximately 20% of the world's population, leading to approximately 3–5 million severe cases and 290,000 to 650,000 deaths each year. Therefore, broad-spectrum potent anti-influenza drugs are needed for seasonal epidemics. The first sentence about 1918 pandemic must be modified and seasonal influenza must be included.
2. Introduction – Please check the number of influenza cases in the US annually. The statement written by authors "The 49 million new cases of influenza have occurred each year in the USA since 2010 ..." raises concerns.
3. Introduction, 2nd paragraph – The authors described only a single class of drugs, e.g. neuraminidase inhibitors, available for the treatment of influenza. However, a new antiviral class called influenza endonuclease inhibitors entered the market in 2018 (Japan, US), represented by baloxavir marboxil (BXM, Xofluza™). BXM targets the influenza A and B virus polymerase acidic protein and inhibits viral gene transcription. The authors must modify Introduction section accordingly.
4. Introduction, 3rd paragraph – The authors must provide more introduction about naturally expressed anti-DNP antibodies. What is the major pathway of developing anti-DNP antibodies? What is lifetime of these antibodies? Why all humans must contain these antibodies?
5. Results, section entitled "The dinitrophenyl conjugate of zanamivir preserves zanamivir's high binding affinity for influenza virus neuraminidases". Here and throughout the text of the manuscript, change "human isoforms of influenza virus" into "different subtypes/lineages of influenza viruses".
6. Results, section entitled "The dinitrophenyl conjugate of zanamivir preserves zanamivir's high binding affinity for influenza virus neuraminidases". Here and throughout the text of the manuscript, change "influenza virus (H1N1)" into "influenza A(H1N1) virus". Change "influenza viruses (H3N2)" into "influenza A(H3N2) viruses".
7. Results, section entitled "zan-DNP is superior to zanamivir in protecting mice from a lethal influenza virus infection". It is recommended to use standard approach for designation of the route of administration of compounds: intranasally administered or intravenously administered. Moreover, it is recommended to use abbreviations for the designation of the route of the drug administration: IN and IM. For example, the quality of Figure 6 will be significantly improved by utilization of these abbreviations for different experimental groups. Some incorrect expressions must be corrected here and throughout the text of the manuscript. For example, it is commonly accepted that mice were

inoculated with influenza virus or lethally challenged with influenza virus. Therefore, change into: "...mice inoculated with a lethal dose of influenza A(H1N1) virus, were treated IN with 0.5 μ mol/kg zan-DNP as described above, only treatment was not initiated until 48, 72 or 96 h after virus inoculation". Please justify why so high virus dose (100 LD50) was used for inoculation of mice? The recommended influenza virus dose (and commonly used) is 3-10 MLD50/mouse. It is recommended to use abbreviation MLD50 (50% mouse lethal dose). For in vivo studies, the doses of drug should be expressed as mg/kg (currently they expressed as μ mol/kg). This will allow comparison with previously published results.

8. Figure 4 – The X axes can be designated as "Days post-infection" and this will correlate with designation of the experimental groups. Moreover, "h post-infection" as designation of the groups can be abbreviated as "hpi", and it will improve the quality of the Figure.

9. Results, section entitled "Systemic administration of zan-DNP is equally potent". Change into: "The fact that high level of protection was demonstrated when neuraminidases from both influenza A and B viruses were targeted, suggests broad-spectrum activity of this antiviral therapy".

10. Figure 5. Expression "systemic administration" raises concerns. It is recommended to use either "parenteral formulation" or "parenteral administration" or "IV administration".

11. Indicate the age, weight and source (country) for female BALB/c mice used in the study.

12. Change into "50% tissue culture infectious dose". Please justify the conditions of TCID50 assay. Why the cells and virus were mixed? Usually monolayers of MDCK cells inoculated with infectious influenza virus.

13. Quantitation of Lung viral titers. In this study, the authors determined virus lung titers by real time RT-PCR. However, determination of infectious virus titers is important.

14. Detection of sera IgG and IgM antibodies to DNP after treatment in mice. Indicate the limit of detection of assay for IgG and IgM antibodies.

Reviewer #2 (Remarks to the Author):

This paper aimed to synthesize and test the efficacy of zanamivir-DNP (zan-DNP), a therapeutic agent that leverages the antiviral activity of neuraminidase inhibitor zanamivir and immunogenicity of the hapten, dinitrophenyl (DNP). The experiments in this paper show that conjugating DNP to zanamivir did not interfere with zanamivir's capacity to bind its target protein neuraminidase, and that zan-DNP was targeted with anti-DNP antibodies both in vitro and in vivo. Utilising a complement-derived cytotoxicity assay and antibody-dependent cytotoxicity assays, the authors also showed that the anti-DNP antibodies kill infected cells in multiple ways. Studies in mice showed that zan-DNP was more effective in treating lethal influenza infection compared to zanamivir alone, even when administered intranasally at 24, 48 h or 72 h post infection. This is an important finding, as a major drawback of NAIs like zanamivir is that treatment is less effective if given 48-hour post-infection. Finally, this study demonstrated that zan-DNP is more effective than zanamivir in mice, when administered through the intraperitoneal route as well.

The design of the compound, zan-DNP, and subsequent experiments in this paper are conceptually sound. The only drawback is that the mouse model is not the best animal model for influenza, and validation of this drug's potency in the ferret model would have provided great insights (PMID: 24709389). However, this reviewer fully understands the prohibitive nature of ferret experiments, due to costs and husbandry requirement. Overall, the findings of this study are novel and significant, and the publication of this paper will be a valuable addition to the literature.

Please see below some specific comments and questions regarding the paper:

Introduction:

'With resistant strains emerging against many of these latter therapies 8-10, new approaches for preventing and treating influenza virus infections are still critically needed'

The term 'resistance' should be replaced with 'viruses with reduced susceptibility'. This statement should be revised altogether to make the point more nuanced. Since the licensure of NAI's such as zanamivir, viruses with reduced susceptibility have only been detected at very low levels (<1%). The only exception was during the 2007-2008 influenza seasons, when seasonal H1N1 viruses with reduced oseltamivir susceptibility circulated widely. Seasonal H1N1 viruses were replaced in 2009/2010 by 'swine flu' (designated H1N1pdm09), which retained sensitivity to oseltamivir. Based on this information, we can make the statement that widespread circulation of viruses with reduced drug susceptibility can occur, rather than that 'resistance is emerging'. This is an important distinction given that data from global susceptibility reports over the last 7 years show consistently low levels of variant viruses with reduced sensitivity. See relevant references below:

Lackenby A, Besselaar TG, Daniels RS, Fry A, Gregory V, Gubareva LV, et al. Global update on the susceptibility of human influenza viruses to neuraminidase inhibitors and status of novel antivirals, 2016-2017. *Antiviral Res.* 2018;157:38-46. Epub 2018/07/10. doi: 10.1016/j.antiviral.2018.07.001. PubMed PMID: 29981793; PubMed Central PMCID: PMC6094047.

Gubareva LV, Besselaar TG, Daniels RS, Fry A, Gregory V, Huang W, et al. Global update on the susceptibility of human influenza viruses to neuraminidase inhibitors, 2015-2016. *Antiviral research.* 2017;146:12-20. doi: 10.1016/j.antiviral.2017.08.004. PubMed PMID: PMC5667636.

Hurt AC, Besselaar TG, Daniels RS, Ermetal B, Fry A, Gubareva L, et al. Global update on the susceptibility of human influenza viruses to neuraminidase inhibitors, 2014-2015. *Antiviral Res.* 2016;132:178-85.

Takashita E, Meijer A, Lackenby A, Gubareva L, Rebelo-de-Andrade H, Besselaar T, et al. Global update on the susceptibility of human influenza viruses to neuraminidase inhibitors, 2013-2014. *Antiviral Res.* 2015;117:27-38.

Meijer A, Rebelo-de-Andrade H, Correia V, Besselaar T, Drager-Dayal R, Fry A, et al. Global update on the susceptibility of human influenza viruses to neuraminidase inhibitors, 2012-2013. *Antiviral Res.* 2014;110:31-41.

Whitley RJ, Boucher CA, Lina B, Nguyen-Van-Tam JS, Osterhaus A, Schutten M, et al. Global Assessment of Resistance to Neuraminidase Inhibitors, 2008-2011: The Influenza Resistance Information Study (IRIS). *Clinical Infectious Diseases.* 2013;56(9):1197-205. doi: 10.1093/cid/cis1220.

Moscona A. Global Transmission of Oseltamivir-Resistant Influenza. *New England Journal of Medicine.* 2009;360(10):953-6. doi: doi:10.1056/NEJMp0900648. PubMed PMID: 19258250.

Results:

The dinitrophenyl conjugate of zanamivir preserves zanamivir's high binding affinity for influenza virus neuraminidases

The Kd data for influenza B viruses, shown in figure 2E and 2F, is several fold higher for zan-DP, compared to zanamivir only. For B/Florida/4/2004 the Kd for zan-DNP is almost 10-folds higher, while for B/Brisbane/60/2008 the Kd for zan-DNP is 8-fold higher, than zanamivir. This is not the case for the influenza A viruses. The authors should comment on this discrepancy in the manuscript.

The EC50 of zan-DNP and zanamivir against influenza A(H1N1) and A(H3N3) was compared but why was influenza B not included in this panel?

zan-DNP is able to recruit anti-DNP antibodies to the surface of virus-infected cells

'To document the ability of zan-DNP to recruit anti-DNP antibodies to virus-infected lung cells in-vivo, we challenged mice with 100x LD50 of A/California/07/2009 (H1N1)pdm09 virus prior to treatment with 0.5 µmol/kg zan-DNP or zanamivir plus anti-DNP antibodies (5 mg/kg, intravenous administration).'

The rationale of including anti-DNP antibodies should be explained earlier in the manuscript.

'Following digestion of the infected lungs, isolate lung cells were evaluated for anti-DNP antibody binding by flow cytometry'

How many days post infection were lungs harvested? This information is not included in the figure legends or methods.

'As shown in the upper left panel of Fig. 3E and Fig. S3, 24.4% of infected lung cells (i.e. hemagglutinin positive cells) were found to be opsonized with antibody in mice treated with zan-DNP. In contrast, replacement of zan-DNP with zanamivir resulted in only 0.8% of the infected lung cells exhibiting antibody positivity (upper right panel), and when uninfected mice were treated with zan-DNP only 1.7% of lung cells were found to be antibody positive. Finally, when zan-DNP was administered to virus-infected mice lacking anti-DNP antibodies, only 0.4% of infected lung cells were determined to be opsonized with antibody. These data demonstrate that significant antibody binding only occurs in vivo when lung cells are exposed to virus, zan-DNP and anti-DNP antibodies, and that absence of any one of these components abrogates antibody binding.'

Given these experiments were performed in triplicates (as shown in Figure S3), it is better to present the values as mean±sd/sem.

Anti-DNP antibodies bound to viral neuraminidase-expressing cells promote infected cell destruction
The 293T cells are described as transfected cells throughout this section. However, in the methods section, it is described that the 293T cells were transduced utilising a lentiviral vector to express neuraminidase (constitutively?). The terms transfected and transduced are not interchangeable, and this section becomes confusing. The terminology should be clarified.

Figure 3F: Some level of cytotoxicity (at least 10%) is also seen with zan-DNP, without anti-DNP antibodies. This should be mentioned.

'Importantly, the bell-shaped response curves shown in Fig. 3G were anticipated, since high concentrations of zan-DNP will begin to inhibit antibody binding after its bridging function has become saturated.'

Please reword for clarity. 'will begin.... become saturated'.

zan-DNP is superior to zanamivir in protecting mice from a lethal influenza virus infection

Figure S9, serum 4 (unimmunized) appears to have similar IgM titres to immunized mice. Was this an outlier?

Figure 4B: There are no error bars for weight loss data from animals treated with zan-DNP 96h post infection.

Systemic administration of zan-DNP is equally potent

Figure 5C and D: Were amounts of zan99mTC in the nose and trachea quantified? From the method it appears that a whole respiratory tract infection of mice was performed. The nose is a site of PR8 virus replication, so it is curious that there is no accumulation of zan-99mTC in 5D. The authors should comment on this.

Moreover, in figure 5D, alongside the kidney, some accumulation of zan-99mTC is seen in the stomach. This should be discussed. Also, how long after infection and drug administration was imaging done?

This is not mentioned in the methods or results.

Legend in figure 5E and the text do not match. Text states that mice received an intraperitoneal injection of 1.5 or 4.5 umol/kg zan-DNP or 4.5 umol/kg of zanamivir. The Figure legends state results from 0.5 or 1.5 umol/kg of zan-DNP and 1.5 umol/kg of zanamivir.

Discussion

In the introduction of this paper, the issue of 'resistance' or viruses with reduced drug susceptibility to NAIs was raised. How does zan-DNP overcome this issue? Viruses have been known to acquire amino acid substitutions, such as R152K(PMID: 9780244), that can reduce binding of its neuraminidase to zanamivir. Presumably, these substitutions would interfere with zan-DNP binding to neuraminidase in the same way they do with zanamivir binding. The authors should discuss this limitation.

Method:

Preparation of influenza virus-infected MDCK cells

What was in the virus growth medium? It should be specified.

CDC and ADC assay: Similar to the results section, the terms transfected and transduced are used interchangeably for 293T cells. This should be corrected.

Quantitation of Lung viral titres:

Reference 12 is not appropriate for the line it is used in.

Supplementary

Supplementary figures and data should appear in the same order as they do in the text.

Reviewer #3 (Remarks to the Author):

In this manuscript, Liu et al describe the use of DNP-coupled zanamivir (zan-DNP) as a potential treatment for influenza infection. They demonstrate that zan-DNP recognizes and binds to influenza virus infected cells in vitro and in vivo. The zanamivir remains enzymatic while anti-DNP antibodies target and kill the infected cells by anti-DNP-induced ADCC and CDC. While the authors induce the anti-DNP antibodies in mice, they speculate that endogenous antibodies in humans will be induced. They do show that humans have anti-DNP antibodies using a small (n=5) cohort. The studies are compelling and novel. This approach will be of interest to the influenza community.

Comments

1. Please address any off-target effects the zan-DNP could have. Is there a potential to target cellular or bacterial NAs/sialidases?
2. It appears that the zan-DNP is effective when administered up to 3 dpi. What advantage does that provide above antiviral alone use?
3. Would zan-DNP work with an NA-resistant mutant virus?
4. What is the cellular role for DNP and anti-DNP antibodies? Is it possible that this approach would result in increased DNP effector or memory B cells that could have a pathologic or autoimmune impact?

Minor

1. No details are provided on the timing of infections in Figures 2, 3 or S2.
2. Please provide more information on the number of biological replicates performed for the in vitro and in vivo studies.
3. The authors immunize extremely young mice (3 weeks of age) to induce anti-DNP. Does the system still work if older mice are used?
4. Figure S5 is not mentioned in the manuscript.

Reviewer #1 (Remarks to the Author):

The hapten-targeted immunotherapies is a rapidly developing approach for killing pathologic cells has already been explored for treatment of many diseases, such as melanomas, prostate cancers and other cancers. Hapten-targeted immunotherapies have also been explored for suppression of activated macrophages in murine models of inflammatory and autoimmune diseases, for therapies against HIV-, bacteria- and fungus-infected cells. In the current study, the authors described the use of zanamivir, a neuraminidase inhibitor of influenza virus, to deliver an attached immunogenic hapten specifically to the surface of virus/virus-infected cells, leading to simultaneous inhibition of virus release and immune-mediated eradication of both free virus and virus-infected cells. This is a novel and very interesting approach in influenza therapy.

MAJOR COMMENTS:

1. Abstract – the composition of a new drug must be clearly described. Abstract requires modifications.

Response: We thank the reviewer for this helpful suggestion. We have added the description of the composition of our new drug to the Abstract. Please see the highlighted changes on page 1.

2. The possible administration routes of zanamivir-DNP should be discussed in more details. The neuraminidase inhibitor zanamivir is still not so often prescribed for influenza due to inhalation route of administration.

Response: We have discussed the possible routes of administration of zanamivir-DNP on page 15 paragraph 2, where it reads: “Second, the route of zan-DNP administration may afford the physician additional treatment options not available with zanamivir. Thus, mice receiving zan-DNP either intranasally or intraperitoneally were found to respond to drug roughly equivalently. If airway congestion or other physical limitations were to found to obstruct administration of zan-DNP intranasally, the physician would have the option to administer the same dose of drug intraperitoneally, subcutaneously or intravenously.”

3. Introduction section requires clarifications about the existing anti-influenza therapies, role of neuraminidase in virus replication cycle, and anti-DNP antibodies.

Response: We thank the reviewer for this suggestion. We have added more information about the current anti-influenza therapies, role of neuraminidase in virus replication cycle and anti-DNP antibodies in the Introduction Section. Please see the highlighted changes on page 1 paragraph 3, and page 2 paragraph 1 & 2.

COMMENTS:

1. Abstract – Seasonal influenza epidemics infect approximately 20% of the world’s population, leading to approximately 3–5 million severe cases and 290,000 to 650,000 deaths each year.

Therefore, broad-spectrum potent anti-influenza drugs are needed for seasonal epidemics. The first sentence about 1918 pandemic must be modified and seasonal influenza must be included.

Response: We thank the reviewer for this good suggestion. We have modified the sentence about 1918 pandemic and included the statistics about seasonal influenza in the Abstract. Please see the highlighted changes on page 1.

2. Introduction – Please check the number of influenza cases in the US annually. The statement written by authors “The 49 million new cases of influenza have occurred each year in the USA since 2010 ...” raises concerns.

Response: We have checked and updated the number of annual influenza cases in the US. We obtained these numbers from the website of the Centers for Disease Control and Prevention (CDC)

(https://www.cdc.gov/flu/about/burden/index.html?CDC_AA_refVal=https%3A%2F%2Fwww.cdc.gov%2Fflu%2Fabout%2Fdisease%2Fus_flu-related_deaths.htm).

We recognized that the statement “The 49 million new cases of influenza have occurred each year in the USA since 2010 ...” may raise concern. Therefore, we have deleted “since 2010” from this statement in order to make it clearer. Please see this highlighted change on page 1, paragraph 2.

3. Introduction, 2nd paragraph – The authors described only a single class of drugs, e.g. neuraminidase inhibitors, available for the treatment of influenza. However, a new antiviral class called influenza endonuclease inhibitors entered the market in 2018 (Japan, US), represented by baloxavir marboxil (BXM, Xofluza™). BXM targets the influenza A and B virus polymerase acidic protein and inhibits viral gene transcription. The authors must modify Introduction section accordingly.

Response: We agree with the reviewer’s suggestion. As we mentioned in our response to the reviewer’s major comments above, we have added the suggested information about baloxavir marboxil to the Introduction section. Please see the highlighted changes on page 2, paragraph 1.

4. Introduction, 3rd paragraph – The authors must provide more introduction about naturally expressed anti-DNP antibodies. What is the major pathway of developing anti-DNP antibodies? What is lifetime of these antibodies? Why all humans must contain these antibodies?

Response: These are also great questions. We have searched for the answer to the function and pathways of induction of anti-DNP antibodies, and unfortunately, while everyone acknowledges that they exist, no one knows why. However, publications indicate that they comprise approximately 1% of circulating antibodies in humans. Speculations on the possible origin of these antibodies have been summarized in a book chapter (McEnaney et al. *ANNU REP MED CHEM*; Vol. 50, p 481) (page 487). We have now cited this reference and added more information on the immunologic properties of the anti-DNP antibodies to the Introduction Section. Please see this highlighted change on page 2, paragraph 2.

5. Results, section entitled “The dinitrophenyl conjugate of zanamivir preserves zanamivir’s high binding affinity for influenza virus neuraminidases”. Here and throughout the text of the

manuscript, change “human isoforms of influenza virus” into “different subtypes/lineages of influenza viruses”.

Response: We agree that the wording “different subtypes/lineages of influenza viruses” is better than “human isoforms of influenza virus”. We have consequently replaced this wording throughout the manuscript.

6. Results, section entitled “The dinitrophenyl conjugate of zanamivir preserves zanamivir’s high binding affinity for influenza virus neuraminidases”. Here and throughout the text of the manuscript, change “influenza virus (H1N1)” into “influenza A(H1N1) virus”. Change “influenza viruses (H3N2)” into “influenza A(H3N2) viruses”.

Response: We have made all of the requested changes throughout the manuscript.

7. Results, section entitled “zan-DNP is superior to zanamivir in protecting mice from a lethal influenza virus infection”. (1) It is recommended to use standard approach for designation of the route of administration of compounds: intranasally administered or intravenously administered. Moreover, it is recommended to use abbreviations for the designation of the route of the drug administration: IN and IM. For example, the quality of Figure 6 will be significantly improved by utilization of these abbreviations for different experimental groups. (2) Some incorrect expressions must be corrected here and throughout the text of the manuscript. For example, it is commonly accepted that mice were inoculated with influenza virus or lethally challenged with influenza virus. Therefore, change into: “...mice inoculated with a lethal dose of influenza A(H1N1) virus, were treated IN with 0.5 $\mu\text{mol/kg}$ zan-DNP as described above, only treatment was not initiated until 48, 72 or 96 h after virus inoculation”. (3) Please justify why so high virus dose (100 LD₅₀) was used for inoculation of mice? The recommended influenza virus dose (and commonly used) is 3-10 MLD₅₀/mouse. It is recommended to use abbreviation MLD₅₀ (50% mouse lethal dose). (4) For in vivo studies, the doses of drug should be expressed as mg/kg (currently they expressed as $\mu\text{mol/kg}$). This will allow comparison with previously published results.

Response:

(1) We have made the requested changes in terminology and abbreviations throughout the text and figures of the manuscript.

(2) We have corrected our wording regarding the route of virus infection. We agree that the words “inoculate” or “challenge” are better than “infuse”. Please see the highlighted changes on page 9, paragraph 1.

(3) We elected to use the high doses (100 LD₅₀) of influenza virus first to clearly demonstrate that zan-DNP is superior to zanamivir and second to document that zan-DNP can hopefully be used to treat patients with very advanced infections (i.e. where other therapies are ineffective). We agree that the abbreviation “MLD₅₀” is better than “LD₅₀”. We have consequently replaced “LD₅₀” with “MLD₅₀” in the manuscript.

(4) We have added a table (Table S1) showing the conversion of units from $\mu\text{mole/kg}$ to mg/kg in supplementary information and cited it in the text of the manuscript when these units first appear.

8. Figure 4 – The X axes can be designated as “Days post-infection” and this will correlate with designation of the experimental groups. Moreover, “h post-infection” as designation of the groups can be abbreviated as “hpi”, and it will improve the quality of the Figure.

Response: We agree with the reviewer’s suggestions. We have made these changes in all figures.

9. Results, section entitled “Systemic administration of zan-DNP is equally potent”. Change into: “The fact that high level of protection was demonstrated when neuraminidases from both influenza A and B viruses were targeted, suggests broad-spectrum activity of this antiviral therapy”.

Response: We have made the recommended change of this sentence in our manuscript. Please see the highlighted change on page 12, paragraph 1.

10. Figure 5. Expression “systemic administration” raises concerns. It is recommended to use either “parenteral formulation” or “parenteral administration” or “IV administration”.

Response: We have replaced “systemic administration” with “parenteral administration” in the text and figures of our manuscript.

11. Indicate the age, weight and source (country) for female BALB/c mice used in the study.

Response: We have added the age, weight and source (country) for female BALB/c mice in the Method Section. Please see the highlighted information on page 22, paragraph 5.

12. Change into “50% tissue culture infectious dose”. Please justify the conditions of TCID50 assay. Why the cells and virus were mixed? Usually monolayers of MDCK cells inoculated with infectious influenza virus.

Response: We recognized that several methods exist for determining the 50% tissue culture infective dose (TCID50) of influenza virus. We performed this assay based on previously published protocols that are cited in the manuscript (Wen et al., *J. Med. Chem.* 52, 4903–4910; Liu et al., *J. Med. Chem.* 55, 8493-8501). The method worked well for our studies.

This colorimetric assay (CellTiter 96® AQueous Non-Radioactive Cell Proliferation Assay) determines the number of viable cells by using non-radioactive reagents to measure the dehydrogenase enzymes that are presented in viable cells. We did try to use monolayers of MDCK cells to perform the above assay. However, we found that when the cells formed monolayers in 96-well plates, the number of cells was too high. As a consequence, even after the cells were incubated with high titer of influenza virus, the amount of dehydrogenase enzymes in the wells of 96-well plates was too high to measure with this assay.

13. Quantitation of Lung viral titers. In this study, the authors determined virus lung titters by real time RT-PCR. However, determination of infectious virus titers is important.

Response: We agree that determination of infectious virus titers is important. There are two reasons why we used real time RT-PCR to determine the titers. First, when we determined the titers of viruses that were harvested from mice, we found that MDCK cells were not susceptible to the mouse-adapted virus (e.g. influenza A virus A/Puerto Rico/8/1934 (H1N1)). As a consequence, the titers of virus determined using MDCK cells were often too low to allow discrimination among different treatment groups. Therefore, we decided to use real time RT-PCR to determine the lung viral titers instead. Second, because we commonly had very large numbers of samples to process, we needed a method like RT-PCR that would enable us to determine the viral titers for all the samples at the same time. In addition, as shown by the figure below, the Cq values determined by real time RT-PCR correlated well with the log (virus dilution). Therefore, we believe that the RT-PCR method provided accurate information on the viral titers.

14. Detection of sera IgG and IgM antibodies to DNP after treatment in mice. Indicate the limit of detection of assay for IgG and IgM antibodies.

Response: We understand that the antibody titer can be defined as the highest dilution that yields an OD (at 450 nm) value above the limit of detection. The detection limit of our ELISA assay was determined as the mean OD of the control groups plus two times the standard deviation of the OD values of the control groups, which were 0.05 and 0.06 for mouse IgG and IgM, respectively. However, because the OD values of mouse anti-DNP antibodies at the highest dilution were still a little higher than the detection limit, we were unable to use the above-mentioned method to define the antibody titer. Instead, we used another commonly used method to define the antibody titer in which the titer was defined as the serum dilution that yielded 50% of the maximal binding on the BSA–DNP-coated plates.

Reviewer #2 (Remarks to the Author):

This paper aimed to synthesize and test the efficacy of zanamivir-DNP (zan-DNP), a therapeutic agent that leverages the antiviral activity of neuraminidase inhibitor zanamivir and immunogenicity of the hapten, dinitrophenyl (DNP). The experiments in this paper show that conjugating DNP to zanamivir did not interfere with zanamivir's capacity to bind its target protein neuraminidase, and that zan-DNP was targeted with anti-DNP antibodies both in vitro and in vivo. Utilising a complement-derived cytotoxicity assay and antibody-dependent cytotoxicity assays, the authors also showed that the anti-DNP antibodies kill infected cells in multiple ways. Studies in mice showed that zan-DNP was more effective in treating lethal influenza infection compared to zanamivir alone, even when administered intranasally at 24, 48 h or 72 h post infection. This is an important finding, as a major drawback of NAIs like zanamivir is that treatment is less effective if given 48-hour post-infection. Finally, this study demonstrated that zan-DNP is more effective than zanamivir in mice, when administered through the intraperitoneal route as well. The design of the compound, zan-DNP, and subsequent experiments in this paper are conceptually sound. The only drawback is that the mouse model is not the best animal model for influenza, and validation of this drug's potency in the ferret model would have provided great insights (PMID: 24709389). However, this reviewer fully understands the prohibitive nature of ferret experiments, due to costs and husbandry requirement. Overall, the findings of this study are novel and significant, and the publication of this paper will be a valuable addition to the literature.

Please see below some specific comments and questions regarding the paper:

Introduction:

'With resistant strains emerging against many of these latter therapies 8-10, new approaches for preventing and treating influenza virus infections are still critically needed' The term 'resistance' should be replaced with 'viruses with reduced susceptibility'. This statement should be revised altogether to make the point more nuanced. Since the licensure of NAI's such as zanamivir, viruses with reduced susceptibility have only been detected at very low levels (<1%). The only exception was during the 2007-2008 influenza seasons, when seasonal H1N1 viruses with reduced oseltamivir susceptibility circulated widely. Seasonal H1N1 viruses were replaced in 2009/2010 by 'swine flu' (designated H1N1pdm09), which retained sensitivity to oseltamivir. Based on this information, we can make the statement that widespread circulation of viruses with reduced drug susceptibility can occur, rather than that 'resistance is emerging'. This is an important distinction given that data from global susceptibility reports over the last 7 years show consistently low levels of variant viruses with reduced sensitivity. See relevant references below:

Lackenby A, Besselaar TG, Daniels RS, Fry A, Gregory V, Gubareva LV, et al. Global update on the susceptibility of human influenza viruses to neuraminidase inhibitors and status of novel

antivirals, 2016-2017. *Antiviral Res.* 2018;157:38-46. Epub 2018/07/10. doi: 10.1016/j.antiviral.2018.07.001. PubMed PMID: 29981793; PubMed Central PMCID: PMC6094047.

Gubareva LV, Besselaar TG, Daniels RS, Fry A, Gregory V, Huang W, et al. Global update on the susceptibility of human influenza viruses to neuraminidase inhibitors, 2015–2016. *Antiviral research.* 2017;146:12-20. doi: 10.1016/j.antiviral.2017.08.004. PubMed PMID: PMC5667636.

Hurt AC, Besselaar TG, Daniels RS, Ermetal B, Fry A, Gubareva L, et al. Global update on the susceptibility of human influenza viruses to neuraminidase inhibitors, 2014-2015. *Antiviral Res.* 2016;132:178-85.

Takashita E, Meijer A, Lackenby A, Gubareva L, Rebelo-de-Andrade H, Besselaar T, et al. Global update on the susceptibility of human influenza viruses to neuraminidase inhibitors, 2013-2014. *Antiviral Res.* 2015;117:27-38.

Meijer A, Rebelo-de-Andrade H, Correia V, Besselaar T, Drager-Dayal R, Fry A, et al. Global update on the susceptibility of human influenza viruses to neuraminidase inhibitors, 2012-2013. *Antiviral Res.* 2014;110:31-41.

Whitley RJ, Boucher CA, Lina B, Nguyen-Van-Tam JS, Osterhaus A, Schutten M, et al. 2013;56(9):1197-205. doi: 10.1093/cid/cis1220.

Moscona A. Global Transmission of Oseltamivir-Resistant Influenza. *New England Journal of Medicine.* 2009;360(10):953-6. doi: doi:10.1056/NEJMp0900648. PubMed PMID: 19258250.

Response: We thank the reviewer for his/her very detailed explanation regarding the prevalence of low levels of variant viruses with reduced susceptibility to neuraminidase inhibitors, and we agree that the expression “virus with reduced susceptibility” is more nuanced than “resistant strains”. We have therefore modified this statement in the Introduction Section (please see the highlighted change on the top of page 2) to reflect the recommended rewording and cite the proposed references.

Results:

The dinitrophenyl conjugate of zanamivir preserves zanamivir’s high binding affinity for influenza virus neuraminidases. The K_d data for influenza B viruses, shown in figure 2E and 2F, is several fold higher for zan-DP, compared to zanamivir only. For B/Florida/4/2004 the K_d for zan-DNP is almost 10-folds higher, while for B/Brisbane/60/2008 the K_d for zan-DNP is 8-fold higher, than zanamivir. This is not the case for the influenza A viruses. The authors should comment on this discrepancy in the manuscript. The EC_{50} of zan-DNP and zanamivir against influenza A(H1N1) and A(H3N3) was compared but why was influenza B not included in this panel?

Response: This is a good question. We recognize that the binding affinity of zan-DNP is 8-10 times lower than zanamivir’s for influenza B viruses (even though it is still in the low nanomolar range). As requested by the reviewer, we have commented in the Result Section on page 3, paragraph 2, that modification of a high affinity inhibitor/ligand with a linker and payload can often lower its affinity for its receptor. We have also cited a review article to support this observation. It is, however, worth mentioning that even though the binding affinity of zan-DNP decreases 10-fold for influenza virus B/Florida/4/2004, it is still potent enough to protect all mice from lethal infections of influenza virus B/Florida/4/2004 (Fig. 4C & 5E).

Regarding the question about why we did not test the EC₅₀ of zan-DNP against influenza B viruses in vitro, because we had already demonstrated that zan-DNP was very effective against influenza A strains in vitro and since zan-DNP was similarly potent against both A and B group viruses in vivo, we did not feel it was important for us to go back and test zan-DNP against influenza B viruses in vitro. Instead, we believe that the binding affinity and mouse therapy studies in vivo are compelling enough to demonstrate the efficacy of our therapy against influenza B virus infections.

zan-DNP is able to recruit anti-DNP antibodies to the surface of virus-infected cells 'To document the ability of zan-DNP to recruit anti-DNP antibodies to virus-infected lung cells in-vivo, we challenged mice with 100x LD50 of A/California/07/2009 (H1N1)pdm09 virus prior to treatment with 0.5 μmol/kg zan-DNP or zanamivir plus anti-DNP antibodies (5 mg/kg, intravenous administration).' The rationale of including anti-DNP antibodies should be explained earlier in the manuscript.

Response: We thank the reviewer for pointing out this issue. We have briefly explained this rationale on page 5, paragraph 5 (see yellow highlighted text).

'Following digestion of the infected lungs, isolate lung cells were evaluated for anti-DNP antibody binding by flow cytometry' How many days post infection were lungs harvested? This information is not included in the figure legends or methods.

Response: We administrated the compounds and antibodies 3 days post-infection and harvested the lungs 12 hours later. Although this information was already included in the Methods Section, we have now added this information to the legend of Fig. 3 to assure that it is readily available when readers are trying to understand the figure. (Please see the highlighted change in the legend of Fig. 3).

'As shown in the upper left panel of Fig. 3E and Fig. S3, 24.4% of infected lung cells (i.e. hemagglutinin positive cells) were found to be opsonized with antibody in mice treated with zan-DNP. In contrast, replacement of zan-DNP with zanamivir resulted in only 0.8% of the infected lung cells exhibiting antibody positivity (upper right panel), and when uninfected mice were treated with zan-DNP only 1.7% of lung cells were found to be antibody positive. Finally, when zan-DNP was administered to virus-infected mice lacking anti-DNP antibodies, only 0.4% of infected lung cells were determined to be opsonized with antibody. These data demonstrate that significant antibody binding only occurs in vivo when lung cells are exposed to virus, zan-DNP and anti-DNP antibodies, and that absence of any one of these components abrogates antibody binding.'

Given these experiments were performed in triplicates (as shown in Figure S3), it is better to present the values as mean±sd/sem.

Response: We thank the reviewer for this good suggestion. We have expanded panel E of Fig. 3 to include a bar graph showing the statistics associated with the flow cytometry study displayed on the left side of panel E and figure S4.

Anti-DNP antibodies bound to viral neuraminidase-expressing cells promote infected cell destruction. The 293T cells are described as transfected cells throughout this section. However, in the methods section, it is described that the 293T cells were transduced utilising a lentiviral vector to express neuraminidase (constitutively?). The terms transfected and transduced are not interchangeable, and this section becomes confusing. The terminology should be clarified.

Response: We agree with the reviewer's comment. The word "transduce" is better since we used the lentiviral vector to introduce foreign DNA into the cells. We have therefore replaced the word "transfect" with "transduce" throughout the manuscript.

Figure 3F: Some level of cytotoxicity (at least 10%) is also seen with zan-DNP, without anti-DNP antibodies. This should be mentioned.

Response: We have added this information to the Results Section. Please see the highlighted change on page 6, paragraph 2.

'Importantly, the bell-shaped response curves shown in Fig. 3G were anticipated, since high concentrations of zan-DNP will begin to inhibit antibody binding after its bridging function has become saturated.' Please reword for clarity. 'will begin.... become saturated'.

Response: we have reworded the sentence to clarify its meaning. Please see the highlighted change on page 6, paragraph 3.

zan-DNP is superior to zanamivir in protecting mice from a lethal influenza virus infection. Figure S9, serum 4 (unimmunized) appears to have similar IgM titres to immunized mice. Was this an outlier?

Response: This was not an outlier. Even after immunization against DNP, the IgM titer of anti-DNP was not significantly higher than the titer in unimmunized mice.

Figure 4B: There are no error bars for weight loss data from animals treated with zan-DNP 96h post infection.

Response: Because only one mouse survived beyond 8 days post-infection, no error bars for weight loss could be provided after eight days. At all previous time points, when multiple mice were still alive, error bars are provided.

systemic administration of zan-DNP is equally potent Figure 5C and D: Were amounts of zan^{99m}Tc in the nose and trachea quantified? From the method it appears that a whole respiratory tract infection of mice was performed. The nose is a site of PR8 virus replication, so it is curious that there is no accumulation of zan-^{99m}Tc in 5D. The authors should comment on this. Moreover, in figure 5D, alongside the kidney, some accumulation of zan-^{99m}Tc is seen in the stomach. This should be discussed. Also, how long after infection and drug administration was imaging done? This is not mentioned in the methods or results.

Response: We did not try to quantify zan-^{99m}Tc in the nose and trachea; we only quantitated uptake in the major organs. We agree that the nose is a site of PR8 virus replication. We do not know why there is no. We assume that the lack of signal of zan-^{99m}Tc in the nose in Fig. 5D is because the number of virus and virus-infected cells in the nose is much lower in total than that in the lungs; i.e. the signal emitted from the nose must be below the detection limit of our SPECT imager. We have added a comment to page 11 paragraph 1 of the manuscript noting this possibility.

We recognized that there is also some radioactivity seen in the large intestine (not stomach) in panel D that was not observed in the quantitation of radioactivity in each organ in panel C. This discrepancy arises because before quantitating the amount of radioactivity in the intestines we washed out the intestinal contents to obtain a true measure of the drug's retention in the actual tissue (i.e. and not in the bowel contents). For prediction of potential off-target toxicities, the amount of drug in the bowel contents will generally have little if any effect.

To further assure the reviewer that zanamivir-targeted conjugates are not accumulating in stomach or intestine, we have attached our near-infrared fluorescence imaging study below (not included in this manuscript). In this study, we use a zanamivir-targeted near IR dye imaging agent (zanamivir-LS288) to study the biodistribution of zanamivir-targeted conjugates. Other than the imaging agent, all the experimental conditions are the same as those used in the above radioimaging study. As shown in both of the images below, there is no zanamivir-LS288 dye present in the stomach or intestine.

Near IR dye-imaging study

Drug group: zanamivir-LS288 (10 nmol)

Competition group: zanamivir-LS288 (10 nmol) + zanamivir (1000 nmol)

In answer to the next question in the above paragraph, we imaged the mice three days post-infection. The images were taken 4 hours after administration of the drugs. This information was already included in the Methods Section. Please see the sentences on page 28, paragraph 2 & 3.

Legend in figure 5E and the text do not match. Text states that mice received an intraperitoneal injection of 1.5 or 4.5 $\mu\text{mol}/\text{kg}$ zan-DNP or 4.5 $\mu\text{mol}/\text{kg}$ of zanamivir. The Figure legends state results from 0.5 or 1.5 $\mu\text{mol}/\text{kg}$ of zan-DNP and 1.5 $\mu\text{mol}/\text{kg}$ of zanamivir.

Response: We thank the reviewer for catching this error. The correct drug doses are 1.5 or 4.5 $\mu\text{mol}/\text{kg}$ zan-DNP and 4.5 $\mu\text{mol}/\text{kg}$ of zanamivir. We have corrected them in the legend of Fig. 5E.

Discussion

In the introduction of this paper, the issue of ‘resistance’ or viruses with reduced drug susceptibility to NAIs was raised. How does zan-DNP overcome this issue? Viruses have been known to acquire amino acid substitutions, such as R152K (PMID: 9780244), that can reduce binding of its neuraminidase to zanamivir. Presumably, these substitutions would interfere with zan-DNP binding to neuraminidase in the same way they do with zanamivir binding. The authors should discuss this limitation.

Response: We agree that if the virus were to mutate its neuraminidase to avoid its binding to zan-DNP, the efficacy of zan-DNP could be affected. However, in light of the high structural similarity between zanamivir and sialic acid (viral neuraminidase’s natural ligand), if the virus were to mutate its ability to bind zanamivir, it would likely simultaneously mutate its ability to hydrolyze sialic acid and therefore compromise its virulence. As a consequence, to date very few mutations have emerged in virulent strains of influenza that have abrogated zanamivir binding.

One possible solution to overcome any possible future drug resistance would be to develop a targeting ligand that could bind a different viral surface protein. As we mentioned in the

Discussion section of our manuscript (page 15 paragraph 3), there are several hemagglutinin inhibitors with nanomolar binding affinities could be developed as targeting ligands for our therapy if zanamivir were to ever fail. We have added a short discussion of this possible limitation and the aforementioned remedy to this paragraph, as suggested by the reviewer.

Method:

Preparation of influenza virus-infected MDCK cells

What was in the virus growth medium? It should be specified.

Response: We prepared the virus growth medium according to the *Manual for the laboratory diagnosis and virological surveillance of influenza*. It is the high glucose DMEM containing 2% BSA, 2% HEPES (1 M), 1% penicillin-streptomycin and 2 µg/ml TPCK-trypsin. We have added this information to the Methods section entitled, "Preparation of influenza virus-infected MDCK cells". Please see the highlighted change on page 23, paragraph 1.

CDC and ADC assay: Similar to the Results section, the terms transfected and transduced are used interchangeably for 293T cells. This should be corrected.

Response: We thank the reviewer for pointing out this issue. As we mentioned in our response to question #6 above. We have replaced the word "transfect" with "transduce" throughout the manuscript.

Quantitation of Lung viral titres:

Reference 12 is not appropriate for the line it is used in.

Response: We checked the manuscript and confirmed that Reference 12 does indeed describe the protocol that we used in this study, where we homogenize the mouse lungs using a gentleMACS Octo Dissociator. Indeed, we employed this protocol to isolate the viral RNA from the infected mouse lungs for the real time RT-PCR study. Therefore, we believe that we should keep this reference in the manuscript. However, we have modified the sentence that cites this reference to render it more understandable. Please see the highlighted change on page 27, paragraph 2.

Supplementary

Supplementary figures and data should appear in the same order as they do in the text.

Response: We have reread the entire manuscript and reordered the supplemental figures.

Reviewer #3 (Remarks to the Author):

In this manuscript, Liu et al describe the use of DNP-coupled zanamivir (zan-DNP) as a potential treatment for influenza infection. They demonstrate that zan-DNP recognizes and binds to influenza virus infected cells in vitro and in vivo. The zanamivir remains enzymatic while anti-DNP antibodies target and kill the infected cells by anti-DNP-induced ADCC and CDC. While the authors induce the anti-DNP antibodies in mice, they speculate that endogenous antibodies in humans will be induced. They do show that humans have anti-DNP antibodies using a small (n=5) cohort. The studies are compelling and novel. This approach will be of interest to the influenza community.

Comments

1. Please address any off-target effects the zan-DNP could have. Is there a potential to target cellular or bacterial NAs/sialidases?

Response: We recognize that some mammals and bacteria express neuraminidases that could conceivably bind zan-DNP and be eliminated by the mechanisms documented here. However, we have concluded that these endogenous neuraminidases should present little if any off-target toxicity problems, since the human neuraminidases bind zanamivir with IC_{50} values that are at least several thousand times weaker than influenza virus neuraminidase (Hata et al., *Antimicrob. Agents Chemother.* 52(10), 3484-3491). This large difference in binding affinity should allow the user to find a concentration of zan-DNP that would bind the influenza virus neuraminidase without binding the endogenous neuraminidase. In case it is of concern, bacterial neuraminidases bind zanamivir with IC_{50} values ranging from 0.1–5 mM (Nishikawa et al., *PLoS ONE* 7(9): e45371); i.e. one million times weaker than influenza virus neuraminidase (IC_{50} ~0.5–3 nM). Moreover, the fact that our biodistribution study in Fig. 5C shows that there is no detectable non-

specific uptake of zanamivir-^{99m}Tc in mice not inoculated with influenza virus (green bars) demonstrates that there is little or no binding of our small molecule conjugate to receptors other than the viral neuraminidase even in mice. We have added a paragraph to address this concern to the Discussion section on page 15 (last paragraph).

2. It appears that the zan-DNP is effective when administered up to 3 dpi. What advantage does that provide above antiviral alone use?

Response: As we mentioned in the manuscript, a major problem with FDA-approved neuraminidase inhibitors like zanamivir is that they are limited to a narrow time frame during which they are effective; i.e. they are only effective during the early stages of the illness, primarily within 24h of infection under the conditions reported here (see Fig. 4B). Since zan-DNP is still effective in protecting infected mice when treatment is delayed up to 3 days post-infection, our therapy has the potential to overcome this significant limitation problem. We believe that this is important, since many patients may not check into a hospital until several days after they begin to notice flu symptoms.

3. Would zan-DNP work with an NA-resistant mutant virus?

Response: Reviewer #2 asked this same question and as we responded to her/him in the response of reviewer 2:

We agree that if the virus were to mutate its neuraminidase to avoid its binding to zan-DNP, the efficacy of zan-DNP could be affected. However, in light of the high structural similarity between zanamivir and sialic acid (viral neuraminidase's natural ligand), if the virus were to mutate its ability to bind zanamivir, it would likely simultaneously mutate its ability to hydrolyze sialic acid and therefore compromise its virulence. As a consequence, to date very few mutations have emerged in virulent strains of influenza that have abrogated zanamivir binding.

One possible solution to overcome any possible future drug resistance would be to develop a targeting ligand that could bind a different viral surface protein. As we mentioned in the Discussion section of our manuscript (page 15 paragraph 3), there are several hemagglutinin inhibitors with nanomolar binding affinities could be developed as targeting ligands for our therapy if zanamivir were to ever fail. We have added a short discussion of this possible limitation and the aforementioned remedy to this paragraph, as suggested by the reviewer.

4. What is the cellular role for DNP and anti-DNP antibodies? Is it possible that this approach would result in increased DNP effector or memory B cells that could have a pathologic or autoimmune impact?

Response: Great question. We have searched for the answer to the function of the anti-DNP antibodies, and no one seems to know. While everyone appears to recognize that anti-DNP antibodies comprise approximately 1% of circulating antibodies in humans, they have no evidence supported hypothesis on why they are produced. Speculations on the possible origin of these antibodies have been summarized in a book chapter (McEnaney et al., *ANNU REP*

MED CHEM; Vol. 50, p 481) (page 487). However, since the anti-DNP antibodies are present in virtually every human, they must be intrinsically nontoxic, arguing that if administration of zan-DNP were to activate DNP effector or memory B cells, no toxicity should emerge. Moreover, given the structure of zan-DNP, it is difficult to imagine how it would be presented to the immune system unless it were to somehow be recognized by a CD1 complex (which is unlikely). And since our zan-DNP conjugate will only promote binding of those antibodies to viral neuraminidase-expressing cells, we anticipate little if any toxicity to uninfected host cells.

Minor

1. No details are provided on the timing of infections in Figures 2, 3 or S2.

Response: We studied the infected MDCK cells in our in vitro assays 24 hours post-infection. We studied the infected NHBE cells 36 hours post-infection. We have added this information to the legends of Figs. 2, 3 and S2. Please see the highlighted changes in these figure legends.

2. Please provide more information on the number of biological replicates performed for the in vitro and in vivo studies.

Response: The number of biological replicates for each experiment were described in the Methods section. We have now added this information to the figure legends.

3. The authors immunize extremely young mice (3 weeks of age) to induce anti-DNP. Does the system still work if older mice are used?

Response: We have conducted related studies using older mice and everything worked very well in these mice also.

4. Figure S5 is not mentioned in the manuscript.

Response: We thank the reviewer for noticing this. We have now cited Figure S5 in the Results section of this manuscript. Please see the highlighted change on page 8, paragraph 2.

REVIEWERS' COMMENTS

Reviewer #1 (Remarks to the Author):

The authors answered reviewers' comments and introduced suggested changes. However, still some changes are required.

COMMENTS:

1. Previous comment "For in vivo studies, the doses of drug should be expressed as mg/kg (currently they expressed as $\mu\text{mol/kg}$). This will allow comparison with previously published results" was addressed by the authors by providing the Supplementary Table 1. This Table 1 shows the conversion of units from $\mu\text{mole/kg}$ to mg/kg. However, this is not what was recommended. Please change all labels and notes to Figures 4 - 6 and Figures S7 - S10 and express the doses in mg/kg. Introduce all the changes into the text of the manuscript.
2. Page 2, 2nd paragraph - Change into "...since zanamivir binds to neuraminidases of all known subtypes/lineages of influenza A and B viruses 6".
3. Page 2, 2nd paragraph - Here and throughout the text of the manuscript, use correct terms. Change into "... group 1 neuraminidase (N1) of influenza A virus...".
4. Page 2, 3rd paragraph - Here and throughout the text of the manuscript, use correct terms. Change into "... of both influenza A(H1N1) and A(H3N2) virus ...". Letter "A" must be included into subtype identification of influenza A viruses.
5. Page 5, 4th paragraph - Change into "... representative member of group 2 neuraminidases of influenza A(H3N2) viruses yielded very similar results (Fig. 3C)."
6. Page 7, figure legend 3 - Change into "DNP antibodies to influenza A(H1N1) (B) and A(H3N2) (D) virus-infected MDCK cells determined...".
7. Page 8, 2nd paragraph - Change into "...vaccinated BALB/c mice against DNP..."
8. Page 8, 2nd paragraph - Change into "in multiple strains of both influenza A and B virus-infected mice...".
9. Figure 4, panel C - Word "dose" is misspelled. Additionally, the authors are using abbreviated designation of PR8 (H1N1) virus and full name of other viruses. It will be better to use full name of all viruses.
10. Figure 5, panel E - Word "dose" is misspelled. It will be better to use full name of all viruses.
11. Figure 6, panels A, B - Word "dose" is misspelled. It will be better to use full name of all viruses.
12. Page 15, 2nd paragraph - Change into "Because zanamivir is believed to treat infections caused by all known subtypes/lineages of influenza A and B viruses 19, ...". Zanamivir cannot treat neuraminidases, it can treat influenza virus infections. Two lineages of influenza B viruses are currently in circulation. There are no subtypes.
13. Page 22, 4th paragraph - Indicate whether mouse-adapted variant of influenza A/California/07/2009 (H1N1)pdm09 virus was used.
14. Page 22, 5th paragraph - the authors wrote "Female BALB/c mice (3 to 4 week old: up to 12g; 6 to 9 week old: 16 to 21g) were purchased from Envigo (location: United States)." Provide the city and state for this company.
15. Figure S2 - Word "dose" is misspelled.
16. Figure S3 - Word "from" is misspelled.

Reviewer #2 (Remarks to the Author):

The manuscript by Liu et al describes the development of Zan-DNP, a small molecule synthesized by conjugating neuraminidase inhibitor zanamivir, to an immunogenic molecule DNP. The authors

demonstrated high effectiveness of this molecule against influenza A and B viruses in cell culture and animal model. A key drawback of this method is that if viruses develop reduced affinity to zanamivir, then the utility of this new therapy would be reduced. However the authors have addressed this in their revision, suggesting HA or M targeting ligands as alternative solution. Overall most of the reviewers comments have been adequately addressed. One minor comment is to refrain from using the word 'cure' throughout the manuscript, as 'cure' refers to symptom alleviation in humans, and outcomes from clinical trials will determine that. It is preferable to say that the drug was shown to be highly effective in reducing viral load in vitro and in vivo.

Reviewer #1 (Remarks to the Author):

The authors answered reviewers' comments and introduced suggested changes. However, still some changes are required.

COMMENTS:

1. Previous comment "For in vivo studies, the doses of drug should be expressed as mg/kg (currently they expressed as $\mu\text{mol/kg}$). This will allow comparison with previously published results" was addressed by the authors by providing the Supplementary Table 1. This Table 1 shows the conversion of units from $\mu\text{mole/kg}$ to mg/kg. However, this is not what was recommended. Please change all labels and notes to Figures 4 - 6 and Figures S7 - S10 and express the doses in mg/kg. Introduce all the changes into the text of the manuscript.

Response: We have now used both units (i.e. $\mu\text{mole/kg}$ and mg/kg) to describe the concentrations used in the figures and text. We believe this addition will satisfy both the biologists and chemists who read the paper.

2. Page 2, 2nd paragraph - Change into "...since zanamivir binds to neuraminidases of all known subtypes/lineages of influenza A and B viruses 6".

Response: We have made the recommended change of this sentence in our manuscript. Please see the highlighted change on page 2, paragraph 2.

3. Page 2, 2nd paragraph - Here and throughout the text of the manuscript, use correct terms. Change into "... group 1 neuraminidase (N1) of influenza A virus...".

Response: We agree with the reviewer's suggestion. We have made all of the requested changes in terminology throughout the manuscript.

4. Page 2, 3rd paragraph – Here and throughout the text of the manuscript, use correct terms. Change into "... of both influenza A(H1N1) and A(H3N2) virus ...". Letter "A" must be included into subtype identification of influenza A viruses.

Response: We agree with the reviewer's suggestion. We have made all of the requested changes in terminology throughout the manuscript.

5. Page 5, 4th paragraph – Change into "... representative member of group 2 neuraminidases of influenza A(H3N2) viruses yielded very similar results (Fig. 3C)."

Response: We have made the recommended change of this sentence in our manuscript. Please see the highlighted change on page 5, paragraph 4.

6. Page 7, figure legend 3 – Change into “DNP antibodies to influenza A(H1N1) (B) and A(H3N2) (D) virus-infected MDCK cells determined...”.

Response: We have made the recommended change of this sentence in our manuscript. Please see the highlighted change on page 7, figure legend 3.

7. Page 8, 2nd paragraph – Change into “...vaccinated BALB/c mice against DNP...”

Response: We have replaced “balb/c mice with “BALB/c mice”. Please see the highlighted change on page 8, paragraph 2.

8. Page 8, 2nd paragraph – Change into “ in multiple strains of both influenza A and B virus-infected mice...”.

Response: We have made the recommended change of this sentence in our manuscript. Please see the highlighted change on page 9, paragraph 2.

9. Figure 4, panel C – Word “dose” is misspelled. Additionally, the authors are using abbreviated designation of PR8 (H1N1) virus and full name of other viruses. It will be better to use full name of all viruses.

Response: We thank the reviewer for catching this error. We have corrected this misspelling in the figure. We have also used full name of all viruses in the figures.

10. Figure 5, panel E – Word “dose” is misspelled. It will be better to use full name of all viruses.

Response: We have corrected this misspelling in the figure. We have also used full name of all viruses in the figure.

11. Figure 6, panels A, B – Word “dose” is misspelled. It will be better to use full name of all viruses.

Response: We have corrected this misspelling in the figure. We have also used full name of all viruses in the figure.

12. Page 15, 2nd paragraph – Change into “Because zanamivir is believed to treat infections caused by all known subtypes/lineages of influenza A and B viruses 19, ...”. Zanamivir cannot treat neuraminidases, it can treat influenza virus infections. Two lineages of influenza B viruses are currently in circulation. There are no subtypes.

Response: We agree with the reviewer's suggestion. We have made the recommended change of this sentence in our manuscript. Please see the highlighted change on page 15, paragraph 2.

13. Page 22, 4th paragraph – Indicate whether mouse-adapted variant of influenza A/California/07/2009 (H1N1)pdm09 virus was used.

Response: We used non-mouse-adapted influenza A/California/07/2009 (H1N1)pdm09 virus. We have added this information to the Methods Section. Please see the highlighted change on page 3, paragraph 4 of Supplementary Information.

14. Page 22, 5th paragraph – the authors wrote “Female BALB/c mice (3 to 4 week old: up to 12g; 6 to 9 week old: 16 to 21g) were purchased from Envigo (location: United States).” Provide the city and state for this company.

Response: We have added the city and state (Indianapolis, IN) for Envigo to the Methods Section. Please see the highlighted change on page 3, paragraph 5 of Supplementary Information.

15. Figure S2 - Word “dose” is misspelled.

Response: We have corrected this misspelling in the figure legend.

16. Figure S3 - Word “from” is misspelled.

Response: We have corrected this misspelling in the figure legend.

Reviewer #2 (Remarks to the Author):

The manuscript by Liu et al describes the development of Zan-DNP, a small molecule synthesized by conjugating neuraminidase inhibitor zanamivir, to an immunogenic molecule DNP. The authors demonstrated high effectiveness of this molecule against influenza A and B viruses in cell culture and animal model. A key drawbacks of this method is that if viruses develop reduced affinity to zanamivir, then the utility of this new therapy would be reduced. However the authors have addressed this in their revision, suggesting HA or M targeting ligands as alternative solution. Overall most of the reviewers comments have been adequately addressed. One minor comment is to refrain from using the word ‘cure’ throughout the manuscript, as ‘cure’ refers to symptom alleviation in humans, and outcomes from clinical trials will determine that. It is preferable to say that the drug was shown to be highly effective in reducing viral load in vitro and in vivo.

Response: We have replaced the word “cure” with several more appropriate terms throughout the manuscript.